# Neural Production Systems

**Aniket Didolkar**[*, 1]**, Anirudh Goyal** [*, 1]**, Nan Rosemary Ke** [2]**, Charles Blundell** [2]**,
Philippe Beaudoin** [3] **Nicolas Heess** [2]**, Michael Mozer** [4, **]**, Yoshua Bengio** [1, **]

## Abstract

Visual environments are structured, consisting of distinct objects or *entities*. These entities have properties—visible or latent—that determine the manner in which they interact with one another. To partition images into entities, deep-learning researchers have proposed structural inductive biases such as slot-based architectures. To model interactions among entities, equivariant graph neural nets (GNNs) are used, but these are not particularly well suited to the task for two reasons. First, GNNs do not predispose interactions to be sparse, as relationships among independent entities are likely to be. Second, GNNs do not factorize knowledge about interactions in an entity-conditional manner. As an alternative, we take inspiration from cognitive science and resurrect a classic approach, *production systems*, which consist of a set of rule templates that are applied by binding placeholder variables in the rules to specific entities. Rules are scored on their match to entities, and the best fitting rules are applied to update entity properties. In a series of experiments, we demonstrate that this architecture achieves a flexible, dynamic flow of control and serves to factorize entity-specific and rule-based information. This disentangling of knowledge achieves robust future-state prediction in rich visual environments, outperforming state-of-the-art methods using GNNs, and allows for the extrapolation from simple (few object) environments to more complex environments.

## 1 Introduction

Despite never having taken a physics course, every child beyond a young age appreciates that pushing a plate off the dining table will cause the plate to break. The laws of physics accurately characterize the dynamics of our natural world, and although explicit knowledge of these laws is not necessary to reason, we can reason explicitly about objects interacting through these laws. Humans can verbalize knowledge in propositional expressions such as "If a plate drops from table height, it will break," and "If a video-game opponent approaches from behind and they are carrying a weapon, they are likely to attack you." Expressing propositional knowledge is not a strength of current deep learning methods for several reasons. First, propositions are discrete and independent from one another. Second, propositions must be quantified in the manner of first-order logic; for example, the video-game proposition applies to any $X$ for which $X$ is an opponent and has a weapon. Incorporating the ability to express and reason about propositions should improve generalization in deep learning methods because this knowledge is modular— propositions can be formulated independently of each other— and can therefore be acquired incrementally. Propositions can also be composed with each other and applied consistently to all entities that match, yielding a powerful form of *systematic generalization*.

The classical AI literature from the 1980s can offer deep learning researchers a valuable perspective. In this era, reasoning, planning, and prediction were handled by architectures that performed propositional inference on symbolic knowledge representations. A simple example of such an architecture is

---
[*] Equal Contribution, [**] Equal Advising [1] Mila, University of Montreal, [2] Google Deepmind, [3] Waverly, [4] Google Research, Brain Team. Corresponding authors: `anirudhgoyal9119@gmail.com`, `adidolkar123@gmail.com`

35th Conference on Neural Information Processing Systems (NeurIPS 2021).

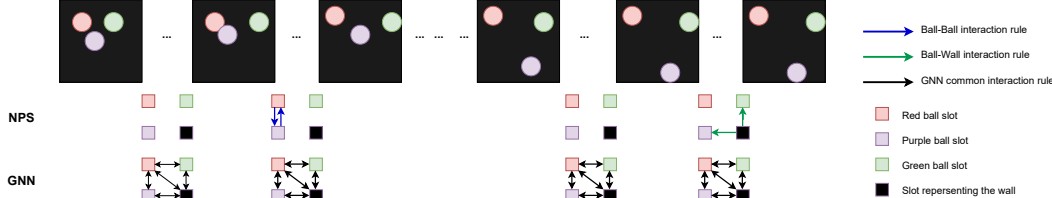

Figure 1: In this figure we show a visual comparison between NPS and dense architectures like GNNs. In NPS, a rule is only applied when an interaction takes place and it is applied only to the slots affected by the interaction. NPS also uses different rules for different kinds of interactions, while in GNN a common rule is applied to all slots irrespective of whether an interaction takes place or not (because of parameter sharing). Note the dynamic nature of the interaction graph in NPS, while in GNN, the graph is static.

the *production system* (Laird et al., 1986; Anderson, 1987), which expresses knowledge by *condition-action rules*. The rules operate on a *working memory*: rule conditions are matched to entities in working memory inspired by cognitive science, and such a match can trigger computational actions that update working memory or external actions that operate on the outside world.

Production systems were typically used to model high-level cognition, e.g., mathematical problem solving or procedure following; perception was not the focus of these models. It was assumed that the results of perception were placed into working memory in a symbolic form that could be operated on with the rules. In this article, we revisit production systems but from a deep learning perspective which naturally integrates perceptual processing and subsequent inference for visual reasoning problems. We describe an end-to-end deep learning model that constructs object-centric representations of entities in videos, and then operates on these entities with differentiable—and thus learnable—production rules. The essence of these rules, carried over from traditional symbolic system, is that they operate on variables that are *bound*, or linked, to the entities in the world. In the deep learning implementation, each production rule is represented by a distinct MLP with query-key attention mechanisms to specify the rule-entity binding and to determine when the rule should be triggered for a given entity. We are not the first to propose a neural instantiation of a production system architecture. Touretzky & Hinton (1988) gave a proof of principle that neural net hardware could be hardwired to implement a production system for symbolic reasoning; our work fundamentally differs from theirs in that (1) we focus on perceptual inference problems and (2) we use the architecture as an inductive bias for learning.

## 1.1 Variables and entities

What makes a rule general-purpose is that it incorporates placeholder *variables* that can be bound to arbitrary *values* or—the term we prefer in this article—*entities*. This notion of binding is familiar in functional programming languages, where these variables are called arguments. Analogously, the use of variables in the production rules we describe enable a model to reason about any set of entities that satisfy the selection criteria of the rule.

Consider a simple function in C like `int add(int a, int b)`. This function binds its two integer operands to variables $a$ and $b$. The function does not apply if the operands are, say, character strings. The use of variables enables a programmer to reuse the same function to add any two integer values

In order for rules to operate on entities, these entities must be represented explicitly. That is, the visual world needs to be parsed in a task-relevant manner, e.g., distinguishing the sprites in a video game or the vehicles and pedestrians approaching an autonomous vehicle. Only in the past few years have deep learning vision researchers developed methods for object-centric representation (Le Roux et al., 2011; Eslami et al., 2016; Greff et al., 2016; Raposo et al., 2017; Van Steenkiste et al., 2018; Kosiorek et al., 2018; Engelcke et al., 2019; Burgess et al., 2019; Greff et al., 2019; Locatello et al., 2020a; Ahmed et al., 2020; Goyal et al., 2019; Zablotskaia et al., 2020; Rahaman et al., 2020; Du et al., 2020; Ding et al., 2020; Goyal et al., 2020; Ke et al., 2021). These methods differ in details but share the notion of a fixed number of *slots* (see Figure 1 for example), also known as *object files*, each encapsulating information about a single object. Importantly, the slots are interchangeable, meaning that it doesn't matter if a scene with an apple and an orange encodes the apple in slot 1 and orange in slot 2 or vice-versa.

A model of visual reasoning must not only be able to represent entities but must also express knowledge about entity dynamics and interactions. To ensure *systematic* predictions, a model must be capable of applying knowledge to an entity regardless of the slot it is in and must be capable of applying the same knowledge to multiple instances of an entity. Several distinct approaches exist in the literature. The predominant approach uses graph neural networks to model slot-to-slot interactions (Scarselli et al., 2008; Bronstein et al., 2017; Watters et al., 2017; Van Steenkiste et al., 2018; Kipf et al., 2018; Battaglia et al., 2018; Tacchetti et al., 2018). To ensure systematicity, the GNN must share parameters among the edges. In a recent article, Goyal et al. (2020) developed a more general framework in which parameters are shared but slots can dynamically select which parameters to use in a state-dependent manner. Each set of parameters is referred to as a *schema*, and slots use a query-key attention mechanism to select which schema to apply at each time step. Multiple slots can select the same schema. In both GNNs and SCOFF, modeling dynamics involves each slot interacting with each other slot. In the work we describe in this article, we replace the direct slot-to-slot interactions with rules, which mediate sparse interactions among slots (See arrows in Figure 1).

Thus our main contribution is that we introduce NPS, which offers a way to model dynamic and sparse interactions among the variables in a graph and also allows dynamic sharing of multiple sets of parameters among these interactions. Most architectures used for modelling interactions in the current literature use statically instantiated graph which model all possible interactions for a given variable at each step i.e. dense interactions. Also such dense architectures share a single set of parameters across all interactions which maybe quite restrictive in terms of representational capacity. A visual comparison between these two kinds of architectures is shown in Figure 1. Through our experiments we show the advantage of modeling interactions in the proposed manner using NPS in visually rich physical environments. We also show that our method results in an intuitive factorization of rules and entities.

## 2   Production System

Formally, our notion of a production system consists of a set of entities and a set of rules, along with a mechanism for selecting rules to apply on subsets of the entities. Implicit in a rule is a specification of the properties of relevant entities, e.g., a rule might apply to one type of sprite in a video game but not another. The control flow of a production system dynamically selects rules as well as bindings between rules and entities, allowing different rules to be chosen and different entities to be manipulated at each point in time.

The neural production system we describe shares essential properties with traditional production system, particularly with regard to the compositionality and generality of the knowledge they embody. Lovett & Anderson (2005) describe four desirable properties commonly attributed to symbolic systems that apply to our work as well.

*Production rules are modular.* Each production rule represents a unit of knowledge and are *atomic* such that any production rule can be intervened (added, modified or deleted) independently of other production rules in the system.

*Production rules are abstract.* Production rules allow for generalization because their conditions may be represented as high-level abstract knowledge that match to a wide range of patterns. These conditions specify the attributes of relationship(s) between entities without specifying the entities themselves. The ability to represent abstract knowledge allows for the transfer of learning across different environments as long as they fit within the conditions of the given production rule.

*Production rules are sparse.* In order that production rules have broad applicability, they involve only a subset of entities. This assumption imposes a strong prior that dependencies among entities are sparse. In the context of visual reasoning, we conjecture that this prior is superior to what has often been assumed in the past, particularly in the disentanglement literature—independence among entities Higgins et al. (2016); Chen et al. (2018).

*Production rules represent causal knowledge and are thus asymmetric.* Each rule can be decomposed into a {condition, action} pair, where the action reflects a state change that is a causal consequence of the conditions being met.

These four properties are sufficient conditions for knowledge to be expressed in production rule form. These properties specify *how* knowledge is represented, but not *what* knowledge is represented. The

latter is inferred by learning mechanisms under the inductive bias provided by the form of production rules.

# 3 Neural Production System: Slots and Sparse Rules

The Neural Production System (NPS), illustrated in Figure 2, provides an architectural backbone that supports the detection and inference of entity (object) representations in an input sequence, and the underlying rules which govern the interactions between these entities in time and space. The input sequence indexed by time step $t$, $\{\boldsymbol{x}^1, \ldots, \boldsymbol{x}^t, \ldots, \boldsymbol{x}^T\}$, for instance the frames in a video, are processed by a neural encoder (Burgess et al., 2019; Greff et al., 2019; Goyal et al., 2019, 2020) applied to each $\boldsymbol{x}^t$, to obtain a set of $M$ entity representations $\{\boldsymbol{V}_1^t, \ldots, \ldots, \boldsymbol{V}_M^t\}$, one for each of the $M$ slots. These representations describe an entity and are updated based on both the previous state, $\boldsymbol{V}^{t-1}$ and the current input, $\boldsymbol{x}^t$.

NPS consists of $N$ separately encoded rules, $\{\boldsymbol{R}_1, \boldsymbol{R}_2, .., \boldsymbol{R}_N\}$. Each rule consists of two components, $\boldsymbol{R}_i = (\vec{\boldsymbol{R}}_i, MLP_i)$, where $\vec{\boldsymbol{R}}_i$ is a learned rule embedding vector, which can be thought of as a template defining the condition for when a rule applies; and $MLP_i$, which determines the action taken by a rule. Both $\vec{\boldsymbol{R}}_i$ and the parameters of $MLP_i$ are learned along with the other parameters of the model using back-propagation on an objective optimized end-to-end.

In the general form of the model, each slot selects a rule that will be applied to it to change its state. This can potentially be performed several times, with possibly different rules applied at each step. Rule selection is done using an attention mechanism described in detail below. Each rule specifies conditions and actions on a pair of slots. Therefore, while modifying the state of a slot using a rule, it can take the state of another slot into account. The slot which is being modified is called the primary slot and other is called the contextual slot. The contextual slot is also selected using an attention mechanism described in detail below.

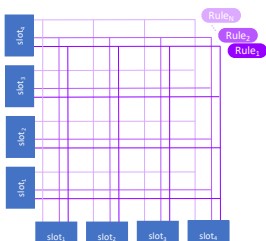

Figure 2: **Rule and slot combinatorics.** Condition-action rules specify how entities interact. Slots maintain the time-varying state of an entity. Every rule is matched to every pair of slots. Through key-value attention, a goodness of match is determined, and a rule is selected along with its binding to slots.

## 3.1 Computational Steps in NPS

In this section, we give a detailed description of the rule selection and application procedure for the slots. First, we will formalize the definitions of a few terms that we will use to explain our method. We use the term **primary slot** to refer to slot $\boldsymbol{V}_p$ whose state gets modified by a rule $\boldsymbol{R}_r$. We use the term **contextual slot** to refer to the slot $\boldsymbol{V}_c$ that the rule $\boldsymbol{R}_r$ takes into account while modifying the state of the primary slot $\boldsymbol{V}_p$.

**Notation.** We consider a set of $N$ rules $\{\boldsymbol{R}_1, \boldsymbol{R}_2, \ldots, \boldsymbol{R}_N\}$ and a set of $T$ input frames $\{\boldsymbol{x}^1, \boldsymbol{x}^2, \ldots, \boldsymbol{x}^T\}$. Each frame $\boldsymbol{x}^t$ is encoded into a set of $M$ slots $\{\boldsymbol{V}_1^t, \boldsymbol{V}_2^t, \ldots, \boldsymbol{V}_M^t\}$. In the following discussion, we omit the index over $t$ for simplicity.

**Step 1.** is external to NPS and involves parsing an input image, $\boldsymbol{x}^t$, into slot-based entities conditioned on the previous state of the slot-based entities. Any of the methods proposed in the literature to obtain a slot-wise representation of entities can be used (Burgess et al., 2019; Greff et al., 2019; Goyal et al., 2019, 2020). The next three steps constitute the rule selection and application procedure.

**Step 2.** For each primary slot $\boldsymbol{V}_p$, we attend to a rule $\boldsymbol{R}_r$ to be applied. Here, the queries come from the primary slot: $\boldsymbol{q}_p = \boldsymbol{V}_p W^q$, and the keys come from the rules: $\boldsymbol{k}_i = \vec{R}_i W^k \quad \forall \boldsymbol{i} \in \{1, \ldots, \boldsymbol{N}\}$. The rule is selected using a straight-through Gumbel softmax (Jang et al., 2016) to achieve a learnable hard decision: $\boldsymbol{r} = \operatorname{argmax}_i(\boldsymbol{q}_p \boldsymbol{k}_i + \gamma)$, where $\gamma \sim \operatorname{Gumbel}(0, 1)$. This competition is a noisy version of rule matching and prioritization in traditional production systems.

**Step 3.** For a given primary slot $\boldsymbol{V}_p$ and selected rule $\boldsymbol{R}_r$, a contextual slot $\boldsymbol{V}_c$ is selected using another attention mechanism. In this case the query comes from the primary slot: $\boldsymbol{q}_p = \boldsymbol{V}_p W^q$, and the keys from all the slots: $\boldsymbol{k}_j = \boldsymbol{V}_j W^q \quad \forall \boldsymbol{j} \in \{1, \ldots, \boldsymbol{M}\}$. The selection takes place using a straight-through Gumbel softmax similar to step 2: $\boldsymbol{c} = \operatorname{argmax}_j(\boldsymbol{q}_p \boldsymbol{k}_j + \gamma)$, where $\gamma \sim \operatorname{Gumbel}(0, 1)$. Note that each rule application is sparse since it takes into account only 1 contextual slot for modifying

a primary slot, while other methods like GNNs take into account all slots for modifying a primary slot.

***Step 4. Rule Application:*** the selected rule $\boldsymbol{R}_r$ is applied to the primary slot $\boldsymbol{V}_p$ based on the rule and the current contents of the primary and contextual slots. The rule-specific $MLP_r$, takes as input the concatenated representation of the state of the primary and contextual slots, $\boldsymbol{V}_p$ and $\boldsymbol{V}_c$, and produces an output, which is then used to change the state of the primary slot $\boldsymbol{V}_p$ by residual addition.

### 3.2   Rule Application: Sequential vs Parallel Rule Application

In the previous section, we have described how each rule application only considers another contextual slot for the given primary slot i.e., **contextual sparsity**. We can also consider **application sparsity**, wherein we use the rules to update the states of only a subset of the slots. In this scenario, only the selected slots would be *primary slots*. This setting will be helpful when there is an entity in an environment that is stationary, or it is following its own default dynamics unaffected by other entities. Therefore, it does not need to consider other entities to update its state. We explore two scenarios for enabling application sparsity.

**Parallel Rule Application.** Each of the $M$ slots selects a rule to potentially change its state. To enable sparse changes, we provide an extra **Null Rule** in addition to the available $N$ rules. If a slot picks the null rule in step 2 of the above procedure, we do not update its state.

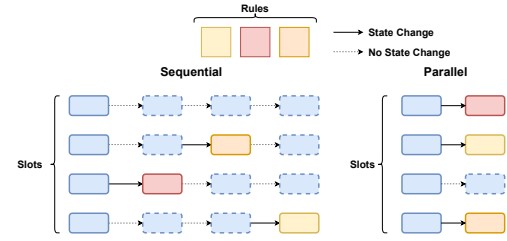

Figure 3: This figure demonstrates the sequential and parallel rule application.

**Sequential Rule Application.** In this setting, only one slot gets updated in each rule application step. Therefore, only one slot is selected as the primary slot. This can be facilitated by modifying step 2 above to select one {primary slot, rule} pair among $\boldsymbol{NM}$ {rule, slot} pairs. The queries come from each slot: $\boldsymbol{q}_j = \boldsymbol{V}_j W^q \quad \forall \boldsymbol{j} \in \{1, \ldots, \boldsymbol{M}\}$, the keys come from the rules: $\boldsymbol{k}_i = R_i W^k \quad \forall \boldsymbol{i} \in \{1, \ldots, \boldsymbol{N}\}$. The straight-through Gumbel softmax selects one (primary slot, rule) pair: $\boldsymbol{p}, \boldsymbol{r} = \mathrm{argmax}_{i,j}(\boldsymbol{q}_p \boldsymbol{k}_i + \gamma)$, where $\gamma \sim \mathrm{Gumbel}(0, 1)$. In the sequential regime, we allow the rule application procedure (step 2, 3, 4 above) to be performed multiple times iteratively in $K$ rule application stages for each time-step $t$.

A pictorial demonstration of both rule application regimes can be found in Figure 3. We provide detailed algorithms for the sequential and parallel regimes in Appendix.

## 4   Experiments

We demonstrate the effectiveness of NPS on multiple tasks and compare to a comprehensive set of baselines. To show that NPS can learn intuitive rules from the data generating distribution, we design a couple of simple toy experiments with well-defined discrete operations. Results show that NPS can accurately recover each operation defined by the data and learn to represent each operation using a separate rule. We then move to a much more complicated and visually rich setting with abstract physical rules and show that factorization of knowledge into rules as offered by NPS does scale up to such settings. We study and compare the parallel and sequential rule application procedures and try to understand the settings which favour each. We then

Table 1: This table shows the segregation of rules for the MNIST Transformation task. Each cell indicates the number of times the corresponding rule is used for the given operation. We can see that NPS automatically and perfectly learns a separate rule for each operation.

|  | Rule 1 | Rule 2 | Rule 3 | Rule 4 |
|---|---|---|---|---|
| Translate Down | 5039 | 0 | 0 | 0 |
| Translate Up | 0 | 4950 | 0 | 0 |
| Rotate Right | 0 | 0 | 5030 | 0 |
| Rotate Left | 0 | 0 | 0 | 4981 |

evaluate the benefits of reusable, dynamic and sparse interactions as offered by NPS in a wide variety of physical environments by comparing it against various baselines. We conduct ablation studies to assess the contribution of different components of NPS. Here we briefly outline the tasks considered and direct the reader to the Appendix for full details on each task and details on hyperparameter settings.

**Discussion of baselines**. NPS is an interaction network, therefore we use other widely used interaction networks such as multihead attention and graph neural networks (Goyal et al. (2019), Goyal et al. (2020), Veerapaneni et al. (2019), Kipf et al. (2019)) for comparison. Goyal et al. (2019) and Goyal

et al. (2020) use an attention based interaction network to capture interactions between the slots, while Veerapaneni et al. (2019) and Kipf et al. (2019) use a GNN based interaction network. We also consider the recently introduced convolutional interaction network (CIN) (Qi et al., 2021) which captures dense pairwise interactions like GNN but uses a convolutional network instead of MLPs to better utilize spatial information. The proposed method, similar to other interaction networks, is agnostic to the encoder backbone used to encode the input image into slots, therefore we compare NPS to other interaction networks across a wide-variety of encoder backbones.

### 4.1 Learning intuitive rules with NPS: Toy Simulations

We designed a couple of simple tasks with well-defined discrete rules to show that NPS can learn intuitive and interpretable rules. We also show the efficiency and effectiveness of the selection procedure (step 2 and step 3 in section 3.1) by comparing against a baseline with many more parameters. Both tasks require a single modification of only one of the available entities, therefore the use of sequential or parallel rule application would not make a difference here since parallel rule application in which all-but-one slots select the null rule is similar to sequential rule application with 1 rule application step. To simplify the presentation, we describe the setup for both tasks using the sequential rule application procedure.

**MNIST Transformation.** We test whether NPS can learn simple rules for performing transformations on MNIST digits. We generate data with four transformations: {Translate Up, Translate Down, Rotate Right, Rotate Left}. We feed the input image ($X$) and the transformation ($o$) to be performed as a one-hot vector to the model. The detailed setup is described in Appendix. For this task, we evaluate whether NPS can learn to use a unique rule for each transformation.

We use 4 rules corresponding to the 4 transformations with the hope that the correct transformations are recovered. Indeed, we observe that **NPS successfully learns to represent each transformation using a separate rule** as shown in Table 1. Our model achieves an MSE of 0.02. A visualization of the outputs from our model and further details can be found in Appendix C.

**Coordinate Arithmetic Task.** The model is tasked with performing arithmetic operations on 2D coordinates. Given $(X_0, Y_0)$ and $(X_1, Y_1)$, we can apply the following operations: {**X Addition**: $(X_r, Y_r) = (X_0 + X_1, Y_0)$, **X Subtraction**: $(X_r, Y_r) = (X_0 - X_1, Y_0)$, **Y Addition**: $(X_r, Y_r) = (X_0, Y_0 + Y_1)$, **Y Subtraction**: $X_r, Y_r = (X_0, Y_0 - Y_1)$}, where $(X_r, Y_r)$ is the resultant coordinate.

In this task, the model is given 2 input coordinates $X = [(x_i, y_i), (x_j, y_j)]$ and the expected output coordinates $Y = [(\hat{x}_i, \hat{y}_i), (\hat{x}_j, \hat{y}_j)]$. The model is supposed to infer the correct rule to produce the correct output coordinates. During data collection, the true output is obtained by performing a random transformation on a randomly selected coordinate in $X$ (primary coordinate), taking another randomly selected coordinate from $X$ (contextual coordinate) into account. The detailed setup is described in Appendix D. We use an NPS model with 4 rules for this task. We use the the selection procedure in step 2 and step 3 of algorithm 1 to select the primary coordinate, contextual coordinate, and the rule. For the baseline we replace the selection procedure in NPS (i.e. step 2 and step 3 in algorithm 1) with a routing MLP similar to Fedus et al. (2021).

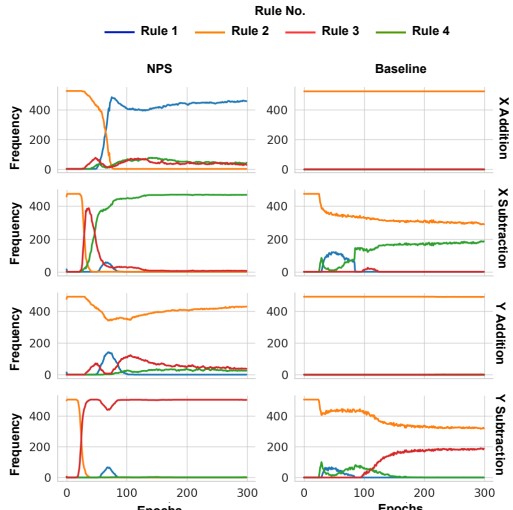

Figure 4: **Coordinate Arithmetic Task.** Here, we compare NPS to the baseline model in terms of segregation of rules as the training progresses. X-axis shows the epochs and Y-axis shows the frequency with which Rule $i$ is used for the given operation. We can see that NPS disentangles the operations perfectly as training progresses with a unique rule specializing to every operation while the baseline model fails to do so.

This routing MLP has 3 heads (one each for selecting the primary coordinate, contextual coordinate, and the rule). The baseline has 4 times more parameters than NPS. The final output is produced by

the rule MLP which does not have access to the true output, hence the model cannot simply copy the true output to produce the actual output. Unlike the MNIST transformation task, we do not provide the operation to be performed as a one-hot vector input to the model, therefore it needs to infer the available operations from the data demonstrations.

We show the segregation of rules for NPS and the baseline in Figure 4. **We can see that NPS learns to use a unique rule for each operation while the baseline struggles to disentangle the underlying operations properly**. **NPS also outperforms the baseline in terms of MSE** achieving an MSE of $0.01_{\pm 0.001}$ while the baseline achieves an MSE of $0.04_{\pm 0.008}$. To further confirm that NPS learns all the available operations correctly from raw data demonstrations, we use an NPS model with 5 rules. **We expect that in this case NPS should utilize only 4 rules since the data describes only 4 unique operations and indeed we observe that NPS ends up mostly utilizing 4 of the available 5 rules** as shown in Table 2.

Table 2: This table shows segregation of rules when we use NPS with 5 rules but the data generation distributions describes only 4 possible operations. We can see that only 4 rules get majorly utilized thus confirming that NPS successfully recovers all possible operations described by the data.

|  | Rule 1 | Rule 2 | Rule 3 | Rule 4 | Rule 5 |
|---|---|---|---|---|---|
| X Addition | 360 | 99 | 45 | 13 | 0 |
| X Subtraction | 0 | 482 | 0 | 1 | 0 |
| Y Subtraction | 0 | 39 | 453 | 2 | 0 |
| Y Addition | 0 | 57 | 15 | 99 | 335 |

### 4.2 Parallel vs Sequential Rule Application

We compare the parallel and sequential rule application procedures, to understand the settings that favour one or the other, over two tasks: (1) Bouncing Balls, (2) Shapes Stack. We use the term PNPS to refer to parallel rule application and SNPS to refer to sequential rule application.

**Shapes Stack.** We use the shapes stack dataset introduced by Groth et al. (2018). This dataset consists of objects stacked on top of each other as shown in Figure 5. These objects fall under the influence of gravity. For our experiments, We follow the same setup as Qi et al. (2021). In this task, given the first frame, the model is tasked with predicting the object bounding boxes for the next $t$ timesteps. The first frame is encoded using a convolutional network followed by RoIPooling (Girshick (2015)) to extract object-centric visual features. The object-centric features are then passed to the dynamics model to predict object bounding boxes of the next $t$ steps. Qi et al. (2021) propose a Region Proposal Interaction Network (RPIN) to solve this task. The dynamics model in RPIN consists of an Interaction Network proposed in Battaglia et al. (2016).

| Model Name | Test | Transfer |
|---|---|---|
| RPIN (Qi et al. (2021)) | $1.254_{\pm 0.008}$ | $6.377_{\pm 0.325}$ |
| PNPS | $1.250_{\pm 0.007}$ | $5.411_{\pm 0.45}$ |
| SNPS | $1.68_{\pm 0.02}$ | $5.80_{\pm 0.15}$ |

Table 3: Prediction error of the compared models on the shapes stack dataset (lower is better) for the test as well as transfer setting. In the test setting the number of rollout steps $t$ is set to 15 and in the transfer setting it is set to 30. We can see that PNPS outperforms the RPIN baseline in both the test and transfer setting while SNPS fails to do so. Results across 15 seeds.

To better utilize spatial information, Qi et al. (2021) propose an extension of the interaction operators in interaction net to operate on 3D tensors. This is achieved by replacing the MLP operations in the original interaction networks with convolutions. They call this new network Convolutional Interaction Network (CIN). For the proposed model, we replace this CIN in RPIN by NPS. To ensure a fair comparison to CIN, we use CNNs to represent rules in NPS instead of MLPs. CIN captures all pairwise interactions between objects using a convolutional network. In NPS, we capture sparse interactions (contextual sparsity) as compared to dense pairwise interactions captured by CIN. Also, in NPS we update only a few subset of slots per step instead of all slots (application sparsity).

We consider two evaluation settings. (1) **Test setting**: The number of rollout timesteps is same as that seen during training (i.e. $t = 15$); (2) **Transfer Setting**: The number of rollout timesteps is higher than that seen during training (i.e. $t = 30$).

We present our results on the shapes stack dataset in Table 3. We can see that both PNPS and SNPS outperform the baseline RPIN in the transfer setting, while only PNPS outperforms the baseline in the test setting and SNPS fails to do so. We can see that PNPS outperforms SNPS. We attribute this to the reduced *application sparsity* with PNPS, i.e., it is more likely that the state of a slot gets updated in PNPS as compared to SNPS. For instance, consider an NPS model with $N$ uniformly chosen rules and $M$ slots. The probability that the state of a slot gets updated in PNPS is $P_{PNPS} = N - 1/N$ (since 1 rule is the null rule), while the same probability for SNPS is $P_{SNPS} = 1/M$ (since only 1 slot gets updated per rule application step).

For this task, we run both PNPS and SNPS for $N = \{1, 2, 4, 6\}$ rules and $M = 3$. For any given $N$, we observe that $P_{PNPS} > P_{SNPS}$. Even when we have multiple rule application steps in SNPS, it might end up selecting the same slot to be updated in more than one of these steps. We report the best performance obtained for PNPS and SNPS across all $N$, which is $N = \{2 + 1 \quad \text{Null Rule}\}$ for PNPS and $N = 4$ for SNPS, in Table 3. Shapes stack is a dataset that would prefer a model with less application sparsity since all the objects are tightly bound to each other (objects are placed on top of each other), therefore all objects spend the majority of their time interacting with the objects directly above or below them. We attribute the higher performance of PNPS compared to RPIN to the higher contextual sparsity of PNPS. Each example in the shapes stack task consists of 3 objects. Even though the blocks are tightly bound to each other, each block is only affected by the objects it is in direct contact with. For example, the top-most object is only affected by the object directly below it. The contextual sparsity offered by PNPS is a strong inductive bias to model such sparse interactions while RPIN models all pairwise interactions between the objects. Figure 5 shows an intuitive illustration of the PNPS model for the shapes stack dataset. In the figure, *Rule 2* actually refers to the Null Rule, while *Rule 1* refers to all the other non-null rules. The bottom-most block picks the Null Rule most times, as the bottom-most block generally does not move.

**Bouncing Balls.** We consider a bouncing-balls environment in which multiple balls move with billiard-ball dynamics. We validate our model on a colored version of this dataset. This is a next-step prediction task in which the model is tasked with predicting the final binary mask of each ball. We compare the following methods: (a) SCOFF (Goyal et al., 2020): factorization of knowledge in terms of slots (object properties) and schemata, the latter capturing object dynamics; (b) SCOFF++: we extend SCOFF by using the idea of iterative competition as proposed in slot attention (SA) (Locatello et al., 2020a); SCOFF + PNPS/SNPS: We replace pairwise slot-to-slot interaction in SCOFF++ with parallel or sequential rule application. For comparing different methods, we use the Adjusted Rand Index or ARI (Rand, 1971). To investigate how the factorization in the form of rules allows for extrapolating knowledge from fewer to more objects, we increase the number of objects from 4 during training to 6-8 during testing.

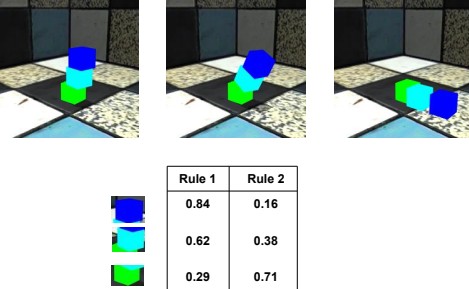

Figure 5: Here we show the rule selection statistics from the proposed model for all entities in the shapes stack dataset across all examples. Each example contains 3 entities as shown above. Each cell in the table shows the probability with which the given rule is triggered for the corresponding entity. We can see that the bottom-most entity triggers rule 2 most of the time while the other 2 entities trigger rule 1 most often. This is quite intuitive as, for most examples, the bottom-most entity remains static and does not move at all while the upper entities fall. Therefore, rule 2 captures information which is relevant to static entities, while rule 1 captures physical rules relevant to the interactions and motion of the upper entities.

We present the results of our experiments in Table 4. Contrary to the shapes stack task, we see that SNPS outperforms PNPS for the bouncing balls task. The balls are not tightly bound together into a single tower as in the shapes stack. Most of the time, a single ball follows its own dynamics, only occasionally interacting with another ball. Rules in NPS capture interaction dynamics between entities, hence they would only be required to change the state of an entity when it interacts with another entity. In the case of bouncing balls, this interaction takes place through a collision between multiple balls. Since for a single ball, such collisions are rare, SNPS, which has higher application sparsity (less probability of modifying the state of an entity), performs better as compared to PNPS (lower application sparsity). Also note that, SNPS has the ability to compose multiple rules together by virtue of having multiple rule application stages. A visualization of the rule and entity selections by the proposed algorithm can be found in Appendix Figure 9.

Given the analysis in this section, we can conclude that PNPS is expected to work better when interactions among entities are more frequent while SNPS is expected to work better when interactions are rare and most of the time, each entity follows its own dynamics. Note that, for both SNPS and PNPS, the rule application considers only 1 other entity as context. Therefore, both approaches have equal *contextual sparsity* while the baselines that we consider (SCOFF and RPIN) capture dense pairwise interactions. We discuss the benefits of *contextual sparsity* in more detail in the next section. More details regarding our setup for the above experiments can be found in Appendix.

## 4.3 Benefits of Sparse Interactions Offered by NPS

In NPS, one can view the computational graph as a dynamically constructed GNN resulting from applying dynamically selected rules, where the states of the slots are represented on the different nodes of the graph, and different rules dynamically instantiate an hyper-edge between a set of slots (the primary slot and the contextual slot). It is important to emphasize that the topology of the graph induced in NPS is dynamic and sparse (only a few nodes affected), while in most GNNs the topology is fixed and dense (all nodes affected). In this section, through a thorough set of experiments, we show that learning sparse and dynamic interactions using NPS indeed works better for the problems we consider than learning dense interactions using GNNs. We consider two types of tasks: (1) Learning Action Conditioned World Models (2) Physical Reasoning. We use SNPS for all these experiments since in the environments that we consider here, interactions among entities are rare.

**Learning Action-Conditioned World Models.** For learning action-conditioned world models, we follow the same experimental setup as Kipf et al. (2019). Therefore, all the tasks in this section are next-$K$ step ($K = \{1, 5, 10\}$) prediction tasks, given the intermediate actions, and with the predictions being performed in the latent space. We use the Hits at Rank 1 (H@1) metrics described by Kipf et al. (2019) for evaluation. H@1 is 1 for a particular example if the predicted state representation is nearest to the encoded true observation and 0 otherwise. We report the average of this score over the test set (higher is better).

| Model Name | Test | Transfer |
|---|---|---|
| SCOFF | 0.28 | 0.15 |
| SCOFF++ | 0.8437 | 0.2632 |
| PNPS (10 Rules+1 Null Rule) | 0.7813 | 0.1997 |
| SNPS (10 Rules) | 0.8518 | 0.3553 |

Table 4: Here we show the ARI achieved by the models on the bouncing balls dataset (higher is better). We can see that SNPS outperforms SCOFF and SCOFF++ while PNPS has a poor performance in this task. Results average across 2 seeds.

**Physics Environment.** The physics environment (Ke et al., 2021) simulates a simple physical world. It consists of blocks of unique but unknown weights. The dynamics for the interaction between blocks is that the movement of heavier blocks pushes lighter blocks on their path. This rule creates an acyclic causal graph between the blocks. For an accurate world model, the learner needs to infer the correct weights through demonstrations. Interactions in this environment are sparse and only involve two blocks at a time, therefore we expect NPS to outperform dense architectures like GNNs. This environment is demonstrated in Appendix Fig 11.

We follow the same setup as Kipf et al. (2019). We use their C-SWM model as baseline. For the proposed model, we only replace the GNN from C-SWM by NPS. GNNs generally share parameters across edges, but in NPS each rule has separate parameters. For a fair comparison to GNN, we use an NPS model with 1 rule. Note that this setting is still different from GNNs as in GNNs at each step every slot is updated by instantiating edges between all pairs of slots, while in NPS an edge is dynamically instantiated between a single pair of slots and only the state of the selected slot (i.e., primary slot) gets updated.

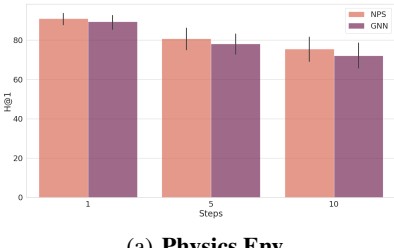

(a) **Physics Env**

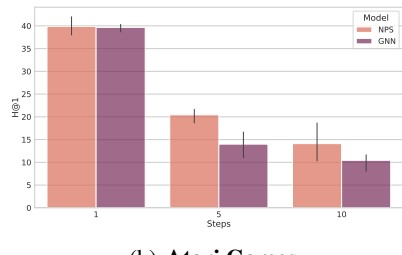

(b) **Atari Games**

Figure 6: **Action-Conditioned World Models**, with number of future steps to be predicted for the world-model on the horizontal axes. (a) Here we show a comparison between GNNs and the proposed NPS on the physics environment using the H@1 metric (higher is better). (b) Comparison of average H@1 scores across 5 Atari games for the proposed model NPS and GNN.

The results of our experiments are presented in Figure 6(a). We can see that NPS outperforms GNNs for all rollouts. Multi-step settings are more difficult to model as errors may get compounded over time steps. The sparsity of NPS (only a single slot affected per step) reduces compounding of errors and enhances symmetry-breaking in the assignment of transformations to rules, while in the

case of GNNs, since all entities are affected per step, there is a higher possibility of errors getting compounded. We can see that even with a single rule, we significantly outperform GNNs thus proving the effectiveness of dynamically instantiating edges between entities.

**Atari Games.** We also test the proposed model in the more complicated setting of Atari. Atari games also have sparse interactions between entities. For instance, in Pong, any interaction involves only 2 entities: (1) paddle and ball or (2) ball and the wall. Therefore, we expect sparse interactions captured by NPS to outperform GNNs here as well.

We follow the same setup as for the physics environment described in the previous section. We present the results for the Atari experiments in Figure 6(b), showing the average H@1 score across 5 games: Pong, Space Invaders, Freeway, Breakout, and QBert. As expected, we can see that the proposed model achieves a higher score than the GNN-based C-SWM. The results for the Atari experiments reinforce the claim that NPS is especially good at learning sparse interactions.

**Learning Rules for Physical Reasoning.** To show the effectiveness of the proposed approach for physical reasoning tasks, we evaluate NPS on another dataset: Sprites-MOT (He et al., 2018). The Sprites-MOT dataset was introduced by He et al. (2018). The dataset contains a set of moving objects of various shapes. This dataset aims to test whether a model can handle occlusions correctly. Each frame has consistent bounding boxes which may cause the objects to appear or disappear from the scene. A model which performs well should be able to track the motion of all objects irrespective of whether they are occluded or not. We follow the same setup as Weis et al. (2020). We use the OP3 model (Veerapaneni et al., 2019) as our baseline. To test the proposed model, we replace the GNN-based transition model in OP3 with the proposed NPS.

We use the same evaluation protocol as followed by Weis et al. (2020) which is based on the MOT (Multi-object tracking) challenge (Milan et al., 2016). The results on the MOTA and MOTP metrics for this task are presented in Table 5. The results on the other metrics are presented in appendix Table 10. We ask the reader to refer to appendix F.1 for more details about these metrics. We can see that for almost all metrics, NPS outperforms the OP3 baseline. Although this dataset does not contain physical interactions between the objects, sparse rule application should still be useful in dealing with occlusions. At any time step, only a single object is affected by occlusions i.e., it may get

| Model | MOTA ↑ | MOTP ↑ |
|-------|--------|--------|
| OP3 | $89.1_{\pm 5.1}$ | $78.4_{\pm 2.4}$ |
| NPS | $90.72_{\pm 5.15}$ | $79.91_{\pm 0.3}$ |

Table 5: **Sprites-MOT**. Comparison between the proposed NPS and the baseline OP3 using the MOTA and MOTP metrics on the sprites-MOT dataset (↑: higher is better). Average over 3 random seeds.

occluded due to another object or due to a prespecified bounding box, while the other objects follow their default dynamics. Therefore, a rule should be applied to only the object (or entity) affected (i.e., not visible) due to occlusion and may take into account any other object or entity that is responsible for the occlusion.

## 5 Discussion and Conclusion

For AI agents such as robots trying to make sense of their environment, the only observables are low-level variables like pixels in images. To generalize well, an agent must induce high-level entities as well as discover and disentangle the rules that govern how these entities actually interact with each other. Here we have focused on perceptual inference problems and proposed NPS, a neural instantiation of production systems by introducing an important inductive bias in the architecture following the proposals of Bengio (2017); Goyal & Bengio (2020); Ke et al. (2021).

**Limitations & Looking Forward.** Our experiments highlight the advantages brought by the factorization of knowledge into a small set of entities and sparse sequentially applied rules. Immediate future work would investigate how to take advantage of these inductive biases for more complex physical environments (Ahmed et al., 2020) and novel planning methods, which might be more sample efficient than standard ones (Schrittwieser et al., 2020).

We also find that Sequential and Parallel NPS have different properties suited towards different domains. Future work should explore how to effectively combine these two approaches. We discuss this in more detail in Appendix section E.3.

# 6    Acknowledgements

The authors would like to thank Matthew Botvinick for useful discussions. The authors would also like to thank Alex Lamb, Stefan Bauer, Nicolas Chapados, Danilo Rezende and Kelsey Allen for brainstorming sessions. We are also thankful to Dianbo Liu, Damjan Kalajdzievski and Osama Ahmed for proofreading. We would like to thank Samsung Electronics Co. Ltd. and CIFAR for funding this research. We would also like to thank Google for providing Google cloud credits used in this work.

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
