# OpenReview forum: "Neural Production Systems"
_NeurIPS.cc/2021/Conference — NeurIPS 2021 Poster_

### Official Review · Reviewer_hBZe · 2021-07-11

**Rating:** 6
**Confidence:** 4

**Summary:**

This paper proposes Neural Production Systems (NPS) which is a (neural) rule-based learning system that operate on explicit entities. The main idea is to extend traditional Production systems to be end-to-end learnable with neural networks. Unlike previous NN systems that model entity-wise interactions such as GNN, NPS allows sparse interactions between entities which is favorable in certain scenarios.

**Limitations And Societal Impact:**

Yes.

**Main Review:**

Originality & Significance
- The idea of extending the traditional rule-based system to neural net rule-based system is intriguing as it can combine the benefit of excellent perception of NN and generalization of rule-based systems. Also, rules in a given environment can be naturally discovered through the learning process. This paper shows promising results on how rules can be learned. However, I have a concern that this paper seems like a marginal improvement over the previously proposed SCOFF (Goyal et al. 2020) model which I further discuss below.

Quality
- SCOFF model also has soft-attention based selection process for entities and hard selection with Gumbel-softmax for schema. Although the intuition and exact formulation is different, the rule application process in NPS can be modelled using schema+soft competition among entities in SCOFF. In that case, the main benefit of NPS comes from having the sparse interaction between entities because it can be more efficient and easier to learn which could be valuable if there is a significant improvement in performance. Therefore, the experiments section should demonstrate that better in my opinion.
- Exp could include more analysis on how NPS can learn certain rules that SCOFF cannot. It would be great to see if the improvements in NPS is enabling the perfect separation of rules in Sec4.1 and if SCOFF cannot do it perfectly.
- The shapes stack and Sec4.3 could also compare with SCOFF as a stronger baseline.
- The internal behavior of NPS is analyzed for simple tasks such as MNIST transformation, but systems like NPS can really shine when applied to complex environments with many entities such as Bouncing Balls. The results would be strengthend if there are detailed analysis or visualization of how entities are selected and how rules are applied to the selected entities.
- Line 402 and Figure 5 are overlapping.

Clarity
- The paper is easy to read and well-written.

**Time Spent Reviewing:**

5

---

> ### Author Response · Authors · 2021-08-10
> **Comparison to SCOFF**
>
> We are glad that the reviewer found the proposed idea intriguing.
>
> **“Relationship With SCOFF”**
>
> We understand that the reviewer is worried about the similarities to SCOFF (Goyal’2020).
> Thanks for giving us the chance to clarify why the benefits of SCOFF and NPS are complementary to each other.
>
> Visual environments are structured, consisting of distinct objects or entities. These entities have properties—visible or latent—that determine the manner in which they interact with one another. To partition images into entities, deep-learning researchers have proposed structural inductive biases such as slot-based architectures. When dealing with representations of structured entities, there are three things that are at play:
>
> - Learning representations of different entities: Factorized in a set of slots.
> - Dynamics of different entities: Consider scenes like a PacMan video-game screen in which the ghosts chase the PacMan, a public square or sports field in which people interact with one another, or a pool table with rolling and colliding balls. In each of these environments, multiple instances of the same object class are present; all operate according to fundamentally similar dynamics. To ensure systematic modeling of the environment, the same dynamics must be applied to multiple object instances, while instances of different objects can follow different dynamics.
> - Interactions between different entities: Different entities need to interact with each other. In GNNs, SCOFF, interactions between different entities are dense (controlled via multihead attention or pairwise interactions). Whereas in NPS, the interactions between different entities are modeled via different rules.
>
> SCOFF deals with the recurrent dynamics of different entities, whereas the NPS deals with the interaction between different entities (sparse in NPS v/s dense in SCOFF). The benefits of NPS and SCOFF are complementary to each other.
>
> **“Analysis on how NPS can learn certain rules that SCOFF cannot”**
>
> We note that in SCOFF, the schemata are learning information about the dynamics about different entities, while NPS involves learning rules which control interactions between different entities.
>
> **"The results would be strengthend if there are detailed analysis or visualization of how entities are selected and how rules are applied to the selected entities."**
>
> We agree with the reviewer. This exactly was the motivation for showing analysis in figure 1 of the paper. Figure 1 shows the rule selection statistics from the proposed model for all entities in the shapes stack dataset across all examples. Each example contains 3 entities as shown above. Each cell in the table shows the probability with which the given rule is triggered for the corresponding entity. We can see that the bottom-most entity triggers rule 2 most of the time while the other 2 entities trigger rule 1 most often. This is quite intuitive as, for most examples, the bottom-most entity remains static and does not move at all while the upper entities fall. Therefore, rule 2 captures information which is relevant to static entities, while rule 1 captures physical rules relevant to the interactions and motion of the upper entities.

---

> > ### Comment · Reviewer_hBZe · 2021-08-22
> > **Thank you for the response**
> >
> > I thank the authors for their response.
> >
> > “Relationship With SCOFF”
> > Yes I agree with the response but as I stated in my review below, I still believe the benefit and difference between NPS and SCOFF has to be demonstrated better in terms of performance or through detailed analysis to strengthen the proposed model.
> > "- In that case, the main benefit of NPS comes from having the sparse interaction between entities because it can be more efficient and easier to learn which could be valuable if there is a significant improvement in performance. Therefore, the experiments section should demonstrate that better in my opinion. - Exp could include more analysis on how NPS can learn certain rules that SCOFF cannot"
> >
> > "The results would be strengthend if there are detailed analysis or visualization of how entities are selected and how rules are applied to the selected entities."
> > In the same vein, as Figure 1 is from a simple environment, I believe it is necessary to analyze rule and entity selection in more environments.

---

> > > ### Author Response · Authors · 2021-08-22
> > > **problems in section 4.1 show that NPS can rule certain rules which SCOFF can't.**
> > >
> > > We thank the reviewer for their reply. We think our experiments already show that NPS can learn certain rules which SCOFF can't.
> > >
> > > **Exp could include more analysis on how NPS can learn certain rules that SCOFF cannot**
> > >
> > > We note that the problems in section 4.1 (MNIST task, and arithmetic task) show that the NPS can learn certain rules which SCOFF can't. For both the MNIST and arithmetic task, we show that the different rules correspond to different arithmetic operations. For such symbolic problems,  we won't be able to use SCOFF to learn such intuitive rules.
> > >
> > > **In that case, the main benefit of NPS comes from having the sparse interaction between entities**
> > >
> > > In NPS, interaction is governed in the form of sparse rules. We have a pool of rules, and each variable/entity can decide which rule to apply. For ex. in GNNs, the norm is to share the parameters across different nodes. NPS provides a way to dynamically share parameters across different nodes. In the case of GNN, it won't be possible to use SCOFF as in SCOFF knowledge is about dynamics of different entities as compared to interactions between different entities.
> > >
> > > **I believe it is necessary to analyze rule and entity selection in more environments**
> > >
> > > We again thank the reviewer. We would be very happy to conduct more experiments.
> > > Are there any concrete problems which the reviewer would recommend for performing more experiments ?

---

> > > > ### Author Response · Authors · 2021-08-24
> > > > **Entity selection for coordinate arithmetic task.**
> > > >
> > > > **it is necessary to analyze rule and entity selection**
> > > >
> > > > As suggested by the reviewer, we further analyze entity selection in NPS for the coordinate arithmetic task.
> > > >
> > > > |    |    1 | 2 |
> > > > | ---- | ---- | ---- |
> > > > | 1  |   76%     |  24%     |
> > > > | 2  |      23%  |   77%    |
> > > >
> > > > In the above table, the rows show the ground truth entity (the entity which should have been selected) and the columns show the selected entity (the entity which was selected by the proposed method). For example, cell [1, 1] shows that in 76% of examples where the 1st entity should have been selected, NPS correctly selects entity 1. We can see that majority of the time (76% in the case of entity 1 and 77% in the case of entity 2), NPS selects the correct entity.

---

> ### Author Response · Authors · 2021-08-28
> **Update Feedback + Experiments showing comparison with SCOFF**
>
> Dear. Reviewer,
>
> We again thank the reviewer for their valuable feedback.
>
> We did more experiments as asked by the reviewer to show the difference between the proposed method and the SCOFF baseline.
> We also did more experiments to visualize the rules learned by the proposed method.
>
> We think that the reviewer feedback was useful and we would like to get feedback on the experiments done by the authors (since the review period is coming to an end).
>
> Thank you for your time and help. We appreciate it. :)

---

### Official Review · Reviewer_CM2F · 2021-07-14

**Rating:** 7
**Confidence:** 4

**Summary:**

This paper proposes a novel method that combines some of the strengths of deep learning with *production systems*, a classic approach in symbolic AI and cognitive science, and evaluates the method on physical reasoning tasks.

**Limitations And Societal Impact:**

Yes, the authors have adequately addressed the limitations and potential negative societal impact of the work.

**Main Review:**

The idea is very exciting and I think it has a lot of promise. Furthermore, the algorithms proposed in the paper creatively combine a range of deep learning techniques (slot-based object-centric representations, key/value attention, gumbel softmax) in an attempt to accomplish this unification. Unfortunately, I think in its current state this work suffers from two limitations:

1) The results:
* The current set of experiments seem to demonstrate only marginal improvements over the baselines in some cases. In many instances there is no specification of how many random seeds are being averaged over, or the number of seeds is very small, and it is not clear whether there is even a statistically significant advantage over the baseline methods.
* Furthermore there is typically only one baseline evaluated for each experiment, and it is not clearly specified why this baseline was chosen or whether it represents the current state-of-the-art for the particular metric being evaluated. This is not the only important consideration, but it is certainly important information to include.

2. The algorithm(s): There are significant questions about the generality of the proposed algorithm, at least in its current formulation.
* First, the algorithm is applied somewhat inconsistently to the different experiments, in a way that isn’t entirely transparent. For instance, in the MNIST transformation task, it sounds like the task instruction (specifying the transformation to be performed) is used to query the rules, which, according to their formulation, should make the task instruction itself the ‘primary slot’, meaning that it should be updated by the selected rule. But presumably it is actually the image embedding that is treated as the primary slot, so as to apply the selected transformation. This experiment is only intended as a preliminary test of the model, but I think it actually reveals a way in which important aspects of the implementation have to be manually tailored to a particular task, because the rule selection and application needs to be handled in an entirely different way if rules are selected on the basis of an instruction vs. on the basis of the objects themselves. Another example comes from the coordinate arithmetic task, in which the coordinates need to be assigned to slots in a particular way in order for the model to successfully use the proposed rule selection method. Specifically, the input and output coordinates for each dimension need to be concatenated into a single slot in order to allow the rule to be inferred from pairwise comparisons between each slot and each rule embedding. If, instead, the input and output coordinates were each represented as separate slots (which seems like a natural way to apply the algorithm as currently specified) there would be no way to identify the rule governing the transformation from input to output.
* Additionally, there are two versions of the proposed method, sequential vs. parallel NPS, and which version works better is somewhat task-dependent. The authors offer reasonable conjectures as to why a particular method works better on a particular task, but this task-dependency does somewhat undermine their claim that sparse interactions are a generally useful inductive bias. The parallel version, which is somewhat less sparse than the sequential version, works better on tasks involving multiple closely interacting objects, whereas the more sparse sequential version works better on tasks involving more spatially distributed objects that mostly don’t interact with each other. But presumably the task of real-world physical reasoning will involve a mixture of these scenarios, and there is no proposal for how this decision should be made other than by the intuitive judgment of the modeler.
* Finally, the current formulation seems to be overly tailored to physical reasoning tasks, and lacking the more general expressive power of the original production systems. For instance, the current formulation only accommodates binary interactions, and there is no way to achieve any kind of hierarchy (i.e. in which the output of one rule serves as the input to another). Many of the design decisions regarding how rules are selected and applied seem as though they have been made specifically with physical reasoning in mind, and it’s not clear how they will be modified to handle other sorts of tasks to which production systems have traditionally been applied, especially tasks involving more abstract reasoning.

In summary, I think the current approach might be more conservatively described as a method for efficiently modeling sparse interactions for the purposes of physical reasoning, rather than as a ‘neural production system’ which seems to imply much more general functionality than it currently has. Nevertheless, I think that it might still merit acceptance if the issues regarding the results and baselines are resolved.

Minor points:
* Some of the notation is inconsistent or unclear. For instance, in the description of step 2 (line 149), the key is obtained using a variable denoted as $R_i$, is this the same as the learned rule embedding vector (described in line 124) denoted as $\vec{\textbf{R}}_i$? Is $\textbf{k}_i$ simply equal to $\vec{\textbf{R}}_i$ as indicated in algorithms 1 and 2, or is it multiplied by the weights $W^k$ as indicated on line 149? In Algorithm 2, are the weights $W^k$ really used to obtain $\textbf{q}_p$ or are these supposed to be $W^q$ or $\widetilde{W}^q$? Also in Algorithm 2, is $\hat{W}^k$ an additional set of weights not specified in the input description, or is it supposed to be $W^k$?
* In the Appendix it says that Table 3 in the paper is wrong and that Table 9 (in the Appendix) reflects the correct results. Hopefully Table 9 will go into the main body of the paper, including the results for both the test and transfer regimes?
* It would be helpful to include pointers to the specific relevant sections in the Appendix, and to try and arrange the sections of the Appendix in the same order as the corresponding sections in the main body of the paper.

**Update after discussion period**

The authors have made improvements to the algorithm, clarified the criteria for selecting baselines and how these relate to the current state-of-the-art, and performed more extensive experiments which demonstrate that the improved algorithm clearly outperforms other approaches on a range of tasks. Notably, these involved experiments on a Raven's Progressive Matrices benchmark, showing some improvements in out-of-distribution generalization, suggesting that NPS may useful in more symbolic domains in addition to the physical reasoning tasks that are the focus of most of the paper. I believe the paper merits acceptance and am updating my score to a 7.


**Time Spent Reviewing:**

6 hours

---

> ### Author Response · Authors · 2021-08-10
> **Regarding Baselines and Results (1/2)**
>
> We are glad that the reviewer found the paper “exciting”.
>
> **Concern regarding experimental results**
>
> The reviewer has raised the concern that our results show only marginal improvements over the baselines and whether our results are statistically significant. We thank the reviewer for pointing this, we made a few minor changes in our NPS model and were able to obtain a much better performance gain. We also show that our results are statistically significant by conducting the t-test. Here we show the new performance on the physics environment and the shapes stack dataset. The changes include the addition of dropout to the attention scores in NPS and using layer normalization after residual addition of the NPS output to the primary slot. We will update the paper with these details and the new results for all other benchmarks as well. Here are the results:
>
>
> Physics Environment:
>
> | Models | H@1 at 1 Step | MRR at 1 Step | H@1 at 5 Step| MRR at 5 Step | H@1 at 10 Step| MRR at 10 |
>
> | NPS    | 91.9 +/- 5.37 | 95.54 +/- 3.01 | 80.82 +/- 8.31 |88.93 +/- 5.31 | 74.28 +/- 11.03 | 84.86 +/- 7.38 |
>
> | C-SWM | 85.8 +/- 9.67 | 91.64 +/- 6.0 | 66.02 +/- 22.13 | 77.83 +/- 17.3 | 54.92 +/- 24.46 | 69.0 +/- 21.67 |
>
> | P-VALUE | 0.15  | 0.15 | 0.10 | 0.10 | 0.06 | 0.07 |
>
> The above results are averaged across 10 seeds, we can see that the p-values are low especially for 5-step and 10-step predictions.
>
> Shapes stack dataset:
>
> | Model | Test | Transfer |
>
> | RPIN | 1.275 +/- 0.009 | 6.471 +/- 0.366 |
>
> | NPS | 1.255 +/- 0.003 | 5.604 +/- 0.589 |
>
> | p-value | 0.64 | 0.09 |
>
> The above results are averaged across 10 seeds. The mean performance of NPS is better for both Test and Transfer settings. We do see a very high p-value for the test setting, but we can see that the p-value for the transfer setting is very low hence showing that the proposed model is definitely superior over longer time scales.
>
> **Regarding Baselines**
>
> NPS is a method for capturing interactions among entities or objects. The main advantage of NPS is as follows:
>
>  - Dynamically instantiating edges in a graph.
>  - Dynamic sharing of parameters among edges. Therefore the parameters of each edge are dynamically selected at run-time from a parameter pool (each rule has separate parameters).
>  - Restricting the number of edges to ensure sparsity.
>
> The proposed experiments aim to show the advantage of each of the above properties. The baselines that we consider are multi-head attention and graph neural networks. In both these baselines, the edges between entities are statically instantiated (i.e. there exists a fixed edge from an entity to every other entity which cannot change), a fixed set of parameters are shared across all these edges and the graph is dense or fully connected (i.e. each entity is connected to every other entity). Our contribution, NPS, is an interaction network, therefore we compare it against other widely used interaction networks such as multi-head attention and graph neural networks ([1], [2], [3], [4]). OP3 [1] and C-SWM [2] use GNN based interaction models, while RIM [3] and SCOFF [4] use MHA based interaction models. Another baseline we consider is the convolutional interaction network [5] which, similar to GNN, captures dense pairwise interaction and has shared parameters across edges. Our method is agnostic to the model used for encoding raw observations into slots.
>
> We would also like to point out that the considered baselines are already shown to be better than other relevant baselines in literature. For example, RPIN [5] is already shown to perform better than baselines such as Object Masking [1] and compositional video prediction [6] as shown in table 1 in [5]. We show that NPS is better than RPIN thus it is better than all models considered in [5].
>
> Similarly, for bouncing balls, SCOFF [4] has already shown superior performance than simple GRU and RIMs [3]. We also compare the performance of NPS to OP3 [1] on bouncing balls (table 10 in supplementary). OP3 is already shown to be better than baselines such as SAVP and O2P2 [1]. We show that NPS outperforms both SCOFF and OP3 both and generalizes much better to the transfer setting with more objects.
>
> For atari, we compare NPS to C-SWM [2]. C-SWM is already shown to be better than monolithic models such as Auto-encoders and Variational auto-encoders in [2].  We show that NPS is better than C-SWM.
>
> To summarize, through our experiments, we have shown that NPS is a better interaction model than other widely used interaction models such as GNNs, MHAs and interaction networks in a wide variety of different backbone encoders which include: OP3 [1], SCOFF[4], C-SWM[2], RPIN [5].
>
>
> [1] Entity Abstraction in Visual Model-Based Reinforcement Learning https://arxiv.org/abs/1910.12827
>
> [2] Contrastive Learning of Structured World Models https://arxiv.org/abs/1911.12247
>
> [3] Recurrent Independent Mechanisms https://arxiv.org/abs/1909.10893
>
> [4] Object Files and Schemata: Factorizing Declarative and Procedural Knowledge in Dynamical Systems https://arxiv.org/abs/2006.16225
>
> [5] Learning Long-Term Visual Dynamics with Region Proposal Interaction Networks https://arxiv.org/abs/2008.02265
>
> [6] Compositional Video Prediction https://arxiv.org/abs/1908.08522

---

> > ### Author Response · Authors · 2021-08-10
> > **About the alogrithm (2/2)**
> >
> > **Regarding Algorithm**
> >
> > We slightly tweak the algorithm in section 4.1 for the mnist and coordinate arithmetic task to highlight different aspects of NPS. As we mentioned before, NPS has 3 properties: (1) Dynamic sharing of parameters among edges using different rules (2) Dynamically instantiating edges among slots (3) Sparsity. In the mnist task, we only want to study the first property i.e. what do the different rules learn? Therefore we fix only one slot to which the rule will be applied (i.e. the image embedding) therefore we don’t need to dynamically instantiate any edge between slots since there is only one slot. In this case, we can see that each rule learns a separate operation. This shows that the learned rules are intuitive. In the coordinate arithmetic task, we want to study whether in addition to rule learning it can instantiate the correct edges among slots. We observe that it does do so. Our remaining experiments analyze all 3 of the above properties together.
> >
> > **Regarding Sequential vs Parallel**
> >
> > The reviewer has raised the concern that the task-dependency of SNPS and PNPS undermines the importance of sparsity. We would like to point out that both SNPS and PNPS have the same contextual sparsity and both consider binary interactions only. Regarding the mixture of these two techniques, we note that how to effectively combine the strengths of the two is an open problem.
> >
> > **“there is no way to achieve any kind of hierarchy (i.e. in which the output of one rule serves as the input to another).”**
> >
> > We thank the reviewer for raising this concern. We do agree that the proposed method is most well-suited to physical reasoning tasks and we leave the extension of this work to other domains as future work. We would like to point out that the SNPS does allow for compositions of rules by virtue of multiple rule application steps.

---

> > > ### Comment · Reviewer_CM2F · 2021-08-11
> > > **Still unclear on status of baselines and results**
> > >
> > > Thanks to the authors for this reply. Here are my responses point-by-point:
> > >
> > > **New results**
> > >
> > > It is good to see that these tweaks to the proposed method yielded improved performance, but it's not clear that this remedies the original concern.
> > > * None of these p values are actually statistically significant (i.e. all p values > 0.05), and the differences between NPS and baseline are generally within the range of error, so it is puzzling that the authors assert otherwise.
> > > * This shows results for the shapes stack and physics environment datasets, but what about the other datasets tested in the paper (atari games, bouncing balls, sprites-MOT)?
> > > * C-SWM appears to perform significantly better in these results than in the original paper, were tweaks also made to this method, and similarly were tweaks made to RPIN relative to what's described in the paper?
> > > * Which version of NPS (parallel or sequential) do these results reflect?
> > > * It appears that one of the entries is missing here for the NPS results on physics environment.
> > >
> > > At this point, it's still unclear whether NPS actually yields a statistically significant improvement over baseline methods.
> > >
> > > **Baselines**
> > >
> > > The authors state that NPS performs better than 'other relevant baselines in the literature', but does this imply that it does not perform better than some other methods, which are for some reason irrelevant? Are these baselines the current state-of-the-art for these datasets? Again, I'd like to emphasize that I don't believe this is the only relevant consideration in determining whether the work is a useful contribution, but it is important to establish how this work relates to the rest of the literature, not only how it relates to other models of the same class (interaction networks).
> > >
> > > Specifically regarding the comparison with RPIN, I took a look at [5] and it looks like the results reported for RPIN there actually outperform both the present implementation of RPIN and NPS, is that the case, and if so how do the authors explain the discrepancy?
> > >
> > > **Algorithm**
> > >
> > > I still believe that the modifications to the algorithm in section 4.1 undermine the apparent generality of the proposed method, at least in its current form. On the surface, the inclusion of these tasks suggest that NPS can be applied to a range of tasks, not just physical reasoning tasks. But in order to be applied to these tasks, fundamental aspects of the algorithm (e.g. how rules are selected and applied, how information is assigned to slots) need to be modified in a task-specific manner, and this isn't clearly spelled out in the paper. In my opinion, it would be better to describe the method more narrowly as a method for modeling sparse interactions in physical reasoning, and to move these other experiments to the Appendix.
> > >
> > > **Minor points**
> > >
> > > Can the authors respond to my questions about notation?

---

> > > > ### Author Response · Authors · 2021-08-17
> > > > **Clarification on Baselines and Results**
> > > >
> > > >
> > > > **Regarding Baselines**
> > > >
> > > > Here we would like to further clarify our choice of baselines. In the paper, we have tested NPS on physical reasoning tasks. Any model applied to physical reasoning tasks has 3 components:
> > > >
> > > > - Representation of different entities i.e. how the raw observations are encoded into slots.
> > > > - Individual dynamics of each entity.
> > > > - Interaction among entities.
> > > >
> > > > Our contribution, NPS, is a contribution to the 3rd component i.e. it is a new kind of interaction network. For the first 2 components, we use the models already available in the literature and those that are prominently used. We refer to the first 2 components as the “backbone”. Since our model is an interaction network, we compare it to other interaction networks used in literature. These include GNNs [1, 2] and Multi-head attention [3, 4]. In every experiment that we perform, we keep the backbone constant and only vary the interaction network (i.e. change it to NPS). In each of our experiments, we show that NPS outperforms other interaction networks while keeping the backbone constant. For example, for bouncing balls, we compare against SCOFF [4]. In the case of SCOFF, we keep the convolutional encoder and the self-dynamics model, which in the case of SCOFF is an RNN, constant and only replace the multi-head attention in SCOFF with NPS. We follow the same procedure for each of our experiments. To ensure that the success of NPS does not depend on any particular backbone, we test it in a number of different backbones which include the OP3 backbone (object masking encoder) for spmot experiments, the RPIN backbone (convolutional encoder followed by convolution self-dynamics model) for shapes stack experiments, SCOFF backbones (slot attention encoder followed by recurrent self-dynamics model) for bouncing balls, etc. In each of these NPS outperforms the default interaction networks used.  In conclusion, the only relevant baselines for NPS are other interaction networks. In deep learning literature, multi-head attention and Graph Neural Networks (interaction networks [5] are similar to GNNs) are generally used as interaction networks.
> > > >
> > > > **Specifically regarding the comparison with RPIN, I took a look at [5] and it looks like the results reported for RPIN there actually outperform both the present implementation of RPIN and NPS, is that the case, and if so how do the authors explain the discrepancy?**
> > > >
> > > > The results reported in [5] are only for 1 seed i.e. the best seed that they got. The authors of [5] have released their best-performing model on their official Github repository (https://github.com/HaozhiQi/RPIN/blob/master/docs/MODEL_ZOO.md) which obtains the exact same performance as reported in the paper. For a fair comparison, we run their code as-is across multiple random seeds and report those results.
> > > >
> > > > **Regarding Results**
> > > >
> > > > We also provide results for atari for 2 games: pong and space invaders. We can see that all our results are statistically significant.
> > > >
> > > > **Pong (across 50 seeds)**
> > > >
> > > > | Model   | H@1 at 1 Step | MRR at 1 Step | H@1 at 5 Step | MRR at 5 Step | H@1 at 10 Step  | MRR at 10 Step |
> > > > | ----- | ---- | ------- | ----- | ----- | ----- | ----- |
> > > > | NPS    |   44.09 +/- 8.30 | 60.577 +/- 7.919 | 22.25 +/- 7.567 | 38.731 +/- 9.005 | 15.615 +/- 5.748 | 30.154 +/- 7.538 |
> > > > | GNN | 30.442 +/- 11.178  |  48.788 +/- 10.428 | 11.596 +/- 6.109 | 25.481 +/- 8.331 | 7.25 +/- 4.057 | 18.942 +/- 6.156 |
> > > > | p-value | 2.75 * 10^-10 | 4.17 * 10^-9 | 4.92 * 10^-12 | 8.48 * 10^-12 | 1.73 * 10^-13 | 6.57 * 10^-13 |
> > > >
> > > > **Space Invaders (across 50 seeds)**
> > > >
> > > > |Model   | H@1 at 1 Step | MRR at 1 Step | H@1 at 5 Step | MRR at 5 Step | H@1 at 10 Step  | MRR at 10 Step |
> > > > | ----- | ---- | ------- | ----- | ----- | ----- | ----- |
> > > > | NPS | 61.269 +/- 11.576 | 74.558 +/- 9.425 | 32.827 +/- 9.179 | 52.981 +/- 9.073 | 21.385 +/- 7.118 | 39.173 +/- 8.505 |
> > > > | GNN | 68.75 +/- 8.44 | 79.942 +/- 5.662 | 21.673 +/- 6.81 | 39.635 +/- 7.395 | 14.712 +/- 4.932 | 31.0 +/- 6.26 |
> > > > |P-value | 0.0003 | 0.0006 | 3.21*10^-10 | 1.002 * 10^-12 | 2.78 * 10^-7 | 2.51 * 10^-7 |
> > > >
> > > > We further run the physics environment experiments for 35 seeds, previously we reported results for 10 seeds. Here we can see that all p-values <= 0.05 except MRR at 10 steps.
> > > >
> > > > **Physics Environment (across 35 seeds)**
> > > >
> > > > | Model   | H@1 at 1 Step | MRR at 1 Step | H@1 at 5 Step | MRR at 5 Step | H@1 at 10 Step  | MRR at 10 Step |
> > > > | ----- | ---- | ------- | ----- | ----- | ----- | ----- |
> > > > | NPS | 93.41 +/-4.33 | 96.255 +/-2.671 | 81.874 +/-11.316 | 89.277 +/-7.532 | 75.615 +/-14.374 | 85.321 +/-9.912 |
> > > > | GNN | 89.397 +/-7.196 | 93.862 +/-4.481 | 76.036 +/-12.71 | 85.4 +/-9.111 | 68.968 +/-15.201 | 80.68 +/-11.45 |
> > > > | P-value | 0.004 | 0.006 | 0.038 | 0.047 | 0.05 | 0.063 |
> > > >
> > > > We further run shapes stack across 15 total seeds, the results we reported before were across 10 seeds. We realize that the performance of NPS on the test set is comparable to RPIN, but on the transfer set, NPS performs much better and is statistically significant.
> > > >
> > > > **Shapes Stack (across 15 seeds)**
> > > >
> > > > | Model | Test | Transfer |
> > > > | ----- | ---- | ------- |
> > > > | RPIN | 1.254 +/- 0.008 | 6.377 +/- 0.325 |
> > > > | NPS | 1.250 +/- 0.007 | 5.411 +/- 0.451 |
> > > > | p-value | 0.89 | 0.0009 |
> > > >
> > > >
> > > > Bouncing balls and sprites-mot being compute-intensive experiments, it was not possible for us to run in the given time-frame (sprites-mot take 20 days on a single V100 GPU).
> > > >
> > > > **C-SWM appears to perform significantly better in these results than in the original paper, were tweaks also made to this method, and similarly were tweaks made to RPIN relative to what's described in the paper?**
> > > >
> > > > We tweaked the backbone for the physics environment experiments by adding Batch normalization to the convolutional encoder hence results in the rebuttal are better than in the original paper. These tweaks were applied to both C-SWM and NPS. For RPIN, we just ran it for more random seeds as compared to the paper.
> > > >
> > > > **Which version of NPS (parallel or sequential) do these results reflect?**
> > > >
> > > > The physics and atari environments experiments reflect sequential NPS and shapes-stack experiments reflect Parallel NPS.
> > > >
> > > > **It appears that one of the entries is missing here for the NPS results on the physics environment.**
> > > >
> > > > We apologize for the error, we have corrected it.
> > > >
> > > > **Still believe in the modifications to the algorithm.**
> > > >
> > > > Let us repeat what the reviewer has said, just to make sure we understand the reviewer’s concern.  Reviewer is concerned that for symbolic tasks  (MNIST and arithmetic tasks), the structure of the proposed model does not exactly follow the algorithm described in the paper, and we need to change some aspects of the algorithm (for ex. how information is assigned to the slots) which makes the reviewer worried about the generality of the method.
> > > >
> > > > We agree with the reviewer’s concern in the way it's described in the paper as of now. That said,  we would like to point out that the primary goal of the symbolic tasks is not to achieve better performance over the baseline, but rather to analyze what different rules are learning. For example, in the arithmetic tasks, different rules correspond to different arithmetic operations. We think that such an intuitive understanding of what different rules are learning is difficult to visually show for other experiments (like world modeling or physical reasoning, as it's hard to “quantify” visual relations for ex. collision). Hence we included such symbolic tasks to give an intuitive understanding to the reader.
> > > >
> > > >
> > > > **Regarding Notation**
> > > >
> > > > - $R_i$ in line 129 is the learned embedding vector $\vec{R_i}$ described in line 124.
> > > > - $k_i$ is obtained by multiplying $\vec{R_i}$ to $W^k$
> > > > - We thank the reviewer for pointing out these discrepancies in algorithm 2. We apologize for the same. $W^q$ should be used to obtain $q_p$ in algorithm 2 instead of $W^k$. $W^k$ should be used to obtain the keys $k_i$. $\hat{W^k}$ should be a separate weight declared in the input.
> > > >
> > > > We apologize for these discrepancies and will correct them in the paper.

---

> > > > > ### Author Response · Authors · 2021-08-19
> > > > > **Stacking Layers of Rules on Top of Each Other**
> > > > >
> > > > > **For instance, the current formulation only accommodates binary interactions, and there is no way to achieve any kind of hierarchy (i.e. in which the output of one rule serves as the input to another)**
> > > > >
> > > > > One way to achieve this would be to combine with Transformers. In Transformers, their are different positions and different positions interact with each other via multi-head attention. NPS is a way to define interactions between different slots/positions. Benefits of NPS and transformer are complementary to each other. In NPS, interactions are mediated via rules, whereas in Transformers interactions are mediated via multi-head dot product attention. The multi-head dot product attention mechanism (MHDPA) runs through the scaled dot-product attention multiple times in parallel. There are two important differences with NPS: in MHDPA, one can treat different heads as different rule applications. Each head or rule considers all the other entities as relevant arguments as compared to the sparse selection of arguments.
> > > > > Another difference is, generally in transformers each layer has a different set of parameters or all the layers share the same set of parameters (like in Universal transformers). In NPS, we just have a pool of rules, and at each position we select a subset of the rules to be applied, and hence this provides a dynamic way to parameter sharing in Transformers.
> > > > >
> > > > > As a preliminary evidence for using NPS in Transformers, we conducted more experiments on the Raven Progressive Matrices benchmark (RPMs) aim to assess analogical reasoning capabilities of deep learning models.
> > > > >
> > > > > **Experimental Setup:** RPMs consist of an incomplete 3 × 3 matrix of context images, and some (typically 8) candidate answer images. The model must decide which of the candidate images is the most appropriate choice to complete the matrix. The content of the panels in a matrix are related by one or more triples, where a triple comprises a logical rule (one of: progression, XOR, OR, AND, consistent union) applied to an attribute (one of: size, type, color, position, number) of an object of a type (one of: shapes, lines). Here, we consider 3 test sets that probe the compositional reasoning ability of the model, namely: Interpolation, Extrapolation, triples.
> > > > >
> > > > > **Training Setup:** Each panel (context and choice) is embedded with a shallow convolutional neural network to obtain an embedding vector. The input to the model is a set of 10 elements, comprising the embeddings of 8 context panels, that of a single candidate panel, and a CLS token (learnable vector). The model output corresponding to the CLS token is fed as input to the prediction head, which is trained to output a score measuring the compatibility of the candidate panel to the context panels. The final prediction is obtained by applying a softmax over the scores of all candidate panels.
> > > > >
> > > > > **Baselines:** We use 6 layered Vision Transformer as the baseline with 4 heads.
> > > > >
> > > > > **Results:** Vision Transformer gets (71.1 +/- 2.1, 19.2 +/- 3.6, 18.9 +/- 2.5) accuracy for (Interpolation, Extrapolation, Held-out triples) (Higher is Better). Proposed architecture (with NPS as interactions) gets (77.6 +/- 1.9 , 28.7 +/- 1.1, 26 +/- 4.2). The hypothesis is that a model which is learning about the data generating distribution should be able to infer and reason about the underlying rule based structure, and hence should be more likely to be successful.
> > > > >
> > > > > We hope that as a preliminary experiment this satisfies the concern of the reviewer.  Future work could investigate how to best integrate the ideas in the proposed work with Transformers to enable hierarchical interactions.

---

> > > > > ### Comment · Reviewer_CM2F · 2021-08-26
> > > > > **Updated results show clear improvement over baselines -- a few remaining issues**
> > > > >
> > > > > The authors have now performed extensive experiments and reported results averaged over a large batch of random seeds, complete with statistics. These new results show that the improved version of the algorithm clearly outperforms the selected baseline in almost all conditions, particularly in settings that emphasize transfer learning. I commend the authors for performing such a thorough evaluation.
> > > > >
> > > > > I do have a few remaining considerations:
> > > > >
> > > > > **Baselines**
> > > > >
> > > > > Thanks to the authors for the clarification regarding selection of baselines. It may be useful to include some version of this description in the revised paper, as it may help readers to better understand how NPS fits into the broader literature on physical reasoning tasks.
> > > > >
> > > > > I agree with the authors contention that the relevant baselines for comparison with NPS are other approaches to modeling interactions. However, I still think it would be useful if the authors can clearly state (both here and in the revised manuscript) whether, to the best of their knowledge, the baselines in question represent the previous state-of-the-art for these tasks.
> > > > >
> > > > > **Modified algorithm for MNIST transformation and arithmetic tasks**
> > > > >
> > > > > The authors argue that the point of these particular experiments is not to show state-of-the-art performance on a difficult task, but to highlight the interpretability of the rules that are learned by the the algorithm. In principle I think it makes sense to use toy tasks for this purpose, but in this case, given that the algorithm has been hand-tailored to these tasks, it arguably does not show that the *proposed algorithm* (i.e. as described in section 2) learns interpretable rules. At the very least, I think it needs to be made clear in the revised manuscript how the algorithm is modified for these tasks, as the impression given in the current version of the paper is that the algorithm can simply be plugged into these more symbolic tasks.
> > > > >
> > > > > **Sequential vs. parallel NPS**
> > > > >
> > > > > As I mentioned in a previous comment, the fact that sequential vs. parallel NPS show different strengths and weaknesses across different datasets does suggest that there's some more work to be done in figuring out how to achieve the right level of sparsity for a given problem. The authors suggest that this is a good topic for future work, which sounds reasonable to me, but I think this should be mentioned in the 'limitations and future work' section.
> > > > >
> > > > > **Experiments on Raven's Progressive Matrices**
> > > > >
> > > > > These experiments were very interesting, and seem to point toward the possibility of NPS being useful in a more symbolic domain. There are a number of details missing from the description provided. What is the CLS token? Which RPM benchmark was investigated here, PGM (there is also the RAVEN benchmark)? The description states that a vision transformer is used as a baseline, but my understanding is that vision transformers operate over image patches, whereas this seems to be operating over embeddings of entire images (embeddings of each cell in the problem). Also from the description it's unclear whether NPS has somehow been integrated into the transformer architecture, or is instead being used as a replacement for the transformer. It would be helpful if the authors could provide as much details as possible about these experiments.
> > > > >
> > > > > The results seem promising, particularly in the transfer ('extrapolation' and 'heldout triples') regimes. I believe that the results in these regimes are better than any I have previously seen. The authors should clarify how these results compare not only to the transformer baseline, but the other previous results on this benchmark.

---

> > > > > > ### Author Response · Authors · 2021-08-26
> > > > > > **Clarifying remaining issues**
> > > > > >
> > > > > > **Baselines**
> > > > > >
> > > > > > **may help readers to better understand how NPS fits into the broader literature on physical reasoning tasks**
> > > > > >
> > > > > > As mentioned by the reviewer we will include more details about selection of the baselines.
> > > > > >
> > > > > > **to the best of their knowledge, the baselines in question represent the previous state-of-the-art for these tasks.**
> > > > > >
> > > > > > To the best of our knowledge, we think our baselines represent the previous state of the art for these tasks.
> > > > > >
> > > > > > **Modified algorithm for MNIST transformation and arithmetic tasks**
> > > > > >
> > > > > > **I think it needs to be made clear in the revised manuscript how the algorithm is modified for these tasks, as the impression given in the current version of the paper is that the algorithm can simply be plugged into these more symbolic tasks**
> > > > > >
> > > > > > We would make it clear that how we modify the algorithm to take into account symbolic tasks.
> > > > > > We would like to point that we do not modify the algorithm in the coordinate arithmetic task, we only change the way in which slots are provided. Otherwise, the {primary slot, rule, contextual slot}  selection is done in exactly the same way as for any other experiment like shapes stack, bouncing balls, or atari.
> > > > > >
> > > > > > **Sequential vs. parallel NPS**
> > > > > >
> > > > > > We will mention the discussion around how to best combine sequential v/s parallel NPS in the 'limitations and future work' section.
> > > > > >
> > > > > >
> > > > > > **Experiments on Raven's Progressive Matrices**
> > > > > >
> > > > > > - We did it on Progressively Generated Matrices (PGM) dataset.
> > > > > > - NPS was added as a replacement of multihead attention in Transformers.
> > > > > > - Each panel (context and choice) is embedded with a convolutional neural network to obtain an embedding vector, and since this is operating on an embedding vector of panels we called this vision transformer, we can just call it normal transformer also. We apologize for the confusion. The CLS token is a learnable vector and the model output corresponding to the CLS token is fed as input to a network which predicts the score over the different context panels measuring the relevance.
> > > > > >
> > > > > > - The experiments reported in the rebuttal were just the proof of concept that the proposed method NPS can be used as an interaction mechanism in Transformers which are already stacked layers of attention (so learning hierarchical representations as asked by the reviewer in their review).
> > > > > >
> > > > > > **The authors have now performed extensive experiments and reported results averaged over a large batch of random seeds, complete with statistics. These new results show that the improved version of the algorithm clearly outperforms the selected baseline in almost all conditions**
> > > > > >
> > > > > > We thank the reviewer for appreciating our efforts in running more experiments and showing that after averaging results on many seeds, the proposed method indeed improves over the various different baselines.
> > > > > >
> > > > > > Since only few days are remaining for the review period, we would be happy to run more toy experiments which can improve our paper.
> > > > > > We even ran more experiments for the manipulation environment (under the heading OP3+NPS outperforms OP3).
> > > > > >
> > > > > > Would the reviewer like to revise their score in the evidence of the extensive experiments done by the authors (and as appreciated by the reviewer) ?
> > > > > >
> > > > > > Thank you for your time, and help. We appreciate it.

---

> > > > > > > ### Comment · Reviewer_CM2F · 2021-08-27
> > > > > > > **Score update**
> > > > > > >
> > > > > > > Thanks to the authors for these additional clarifications. After clarifications concerning the baselines, and additional experiments with improved results, I believe that the paper now merits acceptance. I am updating my score to a 7. I implore the authors to be as transparent as possible in the final revision re: limitations concerning the tradeoffs between sequential vs. parallel NPS, and also to be as clear as possible in describing the modifications to the algorithm for the MNIST experiments. I would also suggest to include the RPM experiments in the main text, and emphasize the improvements in the out-of-distribution generalization regimes, as I think these experiments suggest that the approach may hold promise in more symbolic domains that are traditionally associated with production systems.

---

> > > > > > ### Author Response · Authors · 2021-08-27
> > > > > > **Visualizing rules for bouncing balls**
> > > > > >
> > > > > > We further conduct an analysis of how rules are selected for the bouncing balls task in the paper to better understand the rules learned by NPS. The figure for this analysis can be found here: https://upload.vaa.red/D3rZi#a6138cc7355684ff4d2495495036b405 (shared anonymously). In the figure, we use an NPS model with 3 rules and 3 rule application steps. We analyze the entity selection and the rule selection as asked by the reviewer. A rule application on a slot is shown by highlighting that slot with blue color. The index of the applied rule is also mentioned in the slot. We can see that whenever a rule is applied on the slot representing the background, rule 0 is used. On the other hand, whenever it is applied on the slots representing one of the balls, rule 1 or rule 2 are used. We can also see that rules are mainly only being applied to the two balls in the middle that are touching or close to touching while no rules are being applied to the ball on the top since it stays constant throughout this episode. The ball at the bottom is also mostly constant and receives only 1 rule application when it's close to colliding with the wall. This shows that NPS learns rules that are interpretable in visually complex environments like bouncing balls as well.

---

### Official Review · Reviewer_LeDU · 2021-07-16

**Rating:** 8
**Confidence:** 1

**Summary:**

The paper presents Neural Production Systems (NPS), a neural model to build entiity-centric representations and model the latent rules that determine their interactions in visual environments, inspired by production systems.

**Limitations And Societal Impact:**

the authors adequately addressed the limitations and potential negative societal impact of their work

**Main Review:**

Strengths:
- I found this paper very interesting. Although inspired by an old concept (i.e., production systems), its deep learning adaptation is far from trivial.
- The reported experiments are convincing. In particular, the authors show that NPS can learn arithmetic rules, map rules to MNIST transformation, work in richer visual settings and perform future-state predictions (such as predicting the number of future steps in action-conditioned world models) better than GNNs.
- Rules are differentiable, and could be learned in the context of a broader framework.

Questions:
- How does the model scale with respect to the number of entities/rules considered?

Presentation comments:
- Figure 1 is not colour-blind friendly.
- It might be useful to report results for individual games in Table 5, with the number of entities/rules considered.

**Time Spent Reviewing:**

3

---

> ### Author Response · Authors · 2021-08-10
> **Scaling with respect to number of entities/rules**
>
> We are glad that the reviewer found the paper “very interesting”.
>
>
> **How does the model scale with respect to the number of entities/rules considered?**
>
> That’s an interesting question.  We often found that if we have more slots than the number of objects in the scene, the slot-based architectures would keep some slots “empty” by focussing on low-density visual functions (like background). If we have less number of slots than the number of objects, then the model ignores some objects or even merges two objects into one slot. Similar argument holds for the rules. Rules in NPS are responsible for modelling interactions among different objects. Hence if training distribution contains different numbers of interactions then we often see different rules for different interactions, but if number of interactions are more than number of rules, we notice the same rule being representing multiple interactions. If the number of rules is more than the number of interactions, then we notice some rules are “empty” i.e., they act as no-op operations by keeping the state of the slot similar.
>
> In NPS, Dynamic sharing of parameters as rules scale gracefully as compared to the baselines. In NPS, different rules are parameterized by different sets of  parameters, and the rules are dynamically selected from the parameter pool. For our baselines, there’s parameter sharing across slots to preserve equivariance across the number of slots. Having different parameters for different slots breaks the equivariance across slots, but NPS allows increasing the capacity as well as preserving the equivariance by allowing dynamic weight sharing.
>
>
> **Figure 1 is not colour-blind friendly.**
>
> We apologize for the inconvenience. We would update the figure to make it colour-blind friendly in the next version of the paper.

---

### Official Review · Reviewer_CUnV · 2021-07-17

**Rating:** 6
**Confidence:** 3

**Summary:**

The authors present a new neural network system that can reason over entities. The solution should be applicable to any tasks that require visual reasoning.

Specifically, the NPS algorithm is:
```
for every step in a sequence:
    update the slots (entities)
    compute attention between slot queries and rule keys to select a rule
    compute attention between primary slot queries and all slot keys to get contextual slot
    apply the rule MLP to the primary and contextual slot values
    add the result to the primary slot
```
Slots can either be updated in parallel (i.e., there is only one slot updated per step) or in sequence (i.e., multiple slots can be updated per step).

They compare sequential vs parallel rule application on 4 tasks:
- given a transformation and mnist digit, predict the digit with the transformation applied
- given two x,y coordinates predict the result of an arithmetic change (e.g., subtract the ys from one another) applied
- for a stack of falling shapes, predict the bounding box of each object in future frames
- predict the next frame of billiard balls bouncing around

They compare NPS to GNNs and related methods on four tasks. The goal of each task is to predict latent states in future frames:
- physics env: blocks of unknown weight interacting with one another
- Atari games (pong, space invaders, freeway, breakout, qBert)
- sprites (moving objects of different shapes)

**Ethical Concerns:**

There are no ethical issues.

**Limitations And Societal Impact:**

The authors addressed limitation and impact.

**Main Review:**

Originality: NPS is a novel system for visual reasoning. The computational components borrow heavily from attention mechanisms but separately represent rules to apply.

Quality: The authors test a large number of total tasks, targeting different aspects of NPS. E.g., using the first set of experimental results they are able to provide the recommendation that sequential application is better for sparse interactions.

Clarity: I would have appreciated a more straightforward introduction that was heavier on visuals. E.g., the paragraph beginning at section 64 discusses going from raw observations to slots, which is not what NPS does. It takes up a lot of real estate in the introduction and gave me the impression that the paper focused on going from raw data to slots. It's accompanied by Figure 1, which provides a visualization of rules before they're introduced, which I found confusing. I think it would help with clarity to add more visualizations of the NPS system into the main text (or to at least pull the algorithm into the main text). I also raise some minor inconsistencies that caused confusion under minor comments.

Significance: The potential impact for differentiable reasoning is enormous. The aim of this paper is to take a step towards goal by showing a reasoning system on a variety of restricted settings that are straightforward to analyze. On many tasks, NPS is compared to only one baseline and provides a small performance improvement. E.g., on shapes stack, bouncing balls (test), and sprites, NPS is within 1-2 points of the baseline (and well within the error bars). On physics env and Atari, NPS has a more sizable gain, but is still within the error bars.

Minor comments:
- Algorithm 1 should specify that j indexes M (num slots)
- The steps in algorithm 1 should be consistent with main text (it omits the primary slot selection)
- L208: an -> a?
- L402 figure cuts off the text
- "Entity abstraction in visual model-based reinforcement learning." is cited twice?

Questions to authors:
- how would you contrast NPS to a transformer trained on the same tasks, but with a rule "memory store"?

Related work of interest:
- Visual Grounding of Learned Physical Models. Li et. al, 2020.
- Learning visual predictive models of physics for playing billiards. Fragkiadaki et. al., 2017.

**Time Spent Reviewing:**

6

---

> ### Author Response · Authors · 2021-08-10
> **Adressing Concern Regarding Baselines and Results**
>
> **Clarity**
>
> We thank the reviewer for raising concerns about the clarity of the paper, we will update the paper keeping in mind the given points.
>
> **"On many tasks, NPS is compared to only one baseline"**
>
> NPS is a method for capturing interactions among entities or objects. The main advantage of NPS is as follows:
>
>  - Dynamically instantiating edges in a graph.
>  - Dynamic sharing of parameters among edges. Therefore the parameters of each edge are dynamically selected at run-time from a parameter pool (each rule has separate parameters).
>  - Restricting the number of edges to ensure sparsity.
>
> The proposed experiments aim to show the advantage of each of the above properties. The baselines that we consider are multi-head attention and graph neural networks. In both these baselines, the edges between entities are statically instantiated (i.e. there exists a fixed edge from an entity to every other entity which cannot change), a fixed set of parameters are shared across all these edges and the graph is dense or fully connected (i.e. each entity is connected to every other entity). Our contribution, NPS, is an interaction network, therefore we compare it against other widely used interaction networks such as multi-head attention and graph neural networks ([1], [2], [3], [4]). OP3 [1] and C-SWM [2] use GNN based interaction models, while RIM [3] and SCOFF [4] use MHA based interaction models. Another baseline we consider is the convolutional interaction network [5] which, similar to GNN, captures dense pairwise interaction and has shared parameters across edges. Our method is agnostic to the model used for encoding raw observations into slots.
>
> We would also like to point out that the considered baselines are already shown to be better than other relevant baselines in literature. For example, RPIN [5] is already shown to perform better than baselines such as Object Masking [1] and compositional video prediction [6] as shown in table 1 in [5]. We show that NPS is better than RPIN thus it is better than all models considered in [5].
>
> Similarly, for bouncing balls, SCOFF [4] has already shown superior performance than simple GRU and RIMs [3]. We also compare the performance of NPS to OP3 [1] on bouncing balls (table 10 in supplementary). OP3 is already shown to be better than baselines such as SAVP and O2P2 [1]. We show that NPS outperforms both SCOFF and OP3 both and generalizes much better to the transfer setting with more objects.
>
> For atari, we compare NPS to C-SWM [2]. C-SWM is already shown to be better than monolithic models such as Auto-encoders and Variational auto-encoders in [2].  We show that NPS is better than C-SWM.
>
> To summarize, through our experiments, we have shown that NPS is a better interaction model than other widely used interaction models such as GNNs, MHAs, and interaction networks in a wide variety of different backbone encoders which include: OP3 [1], SCOFF[4], C-SWM[2], RPIN [5].
>
>
> [1] Entity Abstraction in Visual Model-Based Reinforcement Learning https://arxiv.org/abs/1910.12827
>
> [2] Contrastive Learning of Structured World Models https://arxiv.org/abs/1911.12247
>
> [3] Recurrent Independent Mechanisms https://arxiv.org/abs/1909.10893
>
> [4] Object Files and Schemata: Factorizing Declarative and Procedural Knowledge in Dynamical Systems https://arxiv.org/abs/2006.16225
>
> [5] Learning Long-Term Visual Dynamics with Region Proposal Interaction Networks https://arxiv.org/abs/2008.02265
>
> [6] Compositional Video Prediction https://arxiv.org/abs/1908.08522
>
> **On small performance improvement and performance within error bars**
>
> We thank the reviewer for pointing this, we made a few minor changes in our NPS model and were able to obtain a much better performance gain. We also show that our results our statistically significant by conducting the t-test. Here we show the new performance on the physics environment and the shapes stack dataset. The changes include addition of dropout to the attention scores in NPS and using layer normalization after residual addition of the NPS output to the primary slot. We will update the paper with these details and the new results for all other benchmarks as well. Here are the results:
>
>
> Physics Environment:
>
> | Model   | H@1 at 1 Step | MRR at 1 Step | H@1 at 5 Step | MRR at 5 Step | H@1 at 10 Step  | MRR at 10 Step |
> | ----- | ---- | ------- | ----- | ----- | ----- | ----- |
> | NPS | 93.41 +/-4.33 | 96.255 +/-2.671 | 81.874 +/-11.316 | 89.277 +/-7.532 | 75.615 +/-14.374 | 85.321 +/-9.912 |
> | GNN | 89.397 +/-7.196 | 93.862 +/-4.481 | 76.036 +/-12.71 | 85.4 +/-9.111 | 68.968 +/-15.201 | 80.68 +/-11.45 |
> | P-value | 0.004 | 0.006 | 0.038 | 0.047 | 0.05 | 0.063 |
>
> The above results are averaged across 35 seeds, we can see that the p-values are low especially for 5-step and 10-step predictions.
>
> Shapes stack dataset:
>
> | Model | Test | Transfer |
> | ----- | ---- | ------- |
> | RPIN | 1.254 +/- 0.008 | 6.377 +/- 0.325 |
> | NPS | 1.250 +/- 0.007 | 5.411 +/- 0.451 |
> | p-value | 0.89 | 0.0009 |
>
> The above results are averaged across 15 seeds. The mean performance of NPS is better for both Test and Transfer settings. We do see a very high p-value for the test setting, but we can see that the p-value for the transfer setting is very low hence showing that our model is definitely superior over longer time scales.
>
> We also present results for 2 atari games across 50 seeds.
>
> Pong (across 50 seeds):
>
> | Model   | H@1 at 1 Step | MRR at 1 Step | H@1 at 5 Step | MRR at 5 Step | H@1 at 10 Step  | MRR at 10 Step |
> | ----- | ---- | ------- | ----- | ----- | ----- | ----- |
> | NPS    |   44.09 +/- 8.30 | 60.577 +/- 7.919 | 22.25 +/- 7.567 | 38.731 +/- 9.005 | 15.615 +/- 5.748 | 30.154 +/- 7.538 |
> | GNN | 30.442 +/- 11.178  |  48.788 +/- 10.428 | 11.596 +/- 6.109 | 25.481 +/- 8.331 | 7.25 +/- 4.057 | 18.942 +/- 6.156 |
> | p-value | 2.75 * 10^-10 | 4.17 * 10^-9 | 4.92 * 10^-12 | 8.48 * 10^-12 | 1.73 * 10^-13 | 6.57 * 10^-13 |
>
> Space Invaders (across 50 seeds):
>
> |Model   | H@1 at 1 Step | MRR at 1 Step | H@1 at 5 Step | MRR at 5 Step | H@1 at 10 Step  | MRR at 10 Step |
> | ----- | ---- | ------- | ----- | ----- | ----- | ----- |
> | NPS | 61.269 +/- 11.576 | 74.558 +/- 9.425 | 32.827 +/- 9.179 | 52.981 +/- 9.073 | 21.385 +/- 7.118 | 39.173 +/- 8.505 |
> | GNN | 68.75 +/- 8.44 | 79.942 +/- 5.662 | 21.673 +/- 6.81 | 39.635 +/- 7.395 | 14.712 +/- 4.932 | 31.0 +/- 6.26 |
> |P-value | 0.0003 | 0.0006 | 3.21*10^-10 | 1.002 * 10^-12 | 2.78 * 10^-7 | 2.51 * 10^-7 |
>
> **how would you contrast NPS to a transformer trained on the same tasks, but with a rule "memory store"?**
>
> Benefits of NPS and transformer are complementary to each other. In NPS, interactions are mediated via rules, whereas in Transformers interactions are mediated via multi-head dot product attention. The multi-head dot product attention mechanism (MHDPA) runs through the scaled dot-product attention multiple times in parallel. There is an important difference with NPS: in MHDPA, one can treat different heads as different rule applications. Each head (or rule) considers all the other entities as relevant arguments as compared to the sparse selection of arguments in NPS.

---

> > ### Author Response · Authors · 2021-08-19
> > **Preliminary evidence for using NPS in Transformers**
> >
> > As a preliminary evidence for using NPS in Transformers, we conducted more experiments on the Raven Progressive Matrices benchmark (RPMs) aim to assess analogical reasoning capabilities of deep learning models.
> >
> > **Experimental Setup:** RPMs consist of an incomplete 3 × 3 matrix of context images, and some (typically 8) candidate answer images. The model must decide which of the candidate images is the most appropriate choice to complete the matrix. The content of the panels in a matrix are related by one or more triples, where a triple comprises a logical rule (one of: progression, XOR, OR, AND, consistent union) applied to an attribute (one of: size, type, color, position, number) of an object of a type (one of: shapes, lines). Here, we consider 3 test sets  that probe the compositional reasoning ability of the model, namely: Interpolation, Extrapolation, triples.
> >
> > **Training Setup:** Method. Each panel (context and choice) is embedded with a CNN to obtain an embedding vector. The input to the model is a set of 10 elements, comprising the embeddings of 8 context panels, that of a single candidate panel, and a CLS token (learnable vector). The model output corresponding to the CLS token is fed as input to the prediction head, which is trained to output a score measuring the compatibility of the candidate panel to the context panels. The final prediction is obtained by applying a softmax over the scores of all candidate panels.
> >
> > **Baselines:** We use 6 layered Vision Transformer as the baseline with 4 attention heads.
> >
> > **Integration with NPS:** Benefits of NPS and transformer are complementary to each other. In NPS, interactions are mediated via rules, whereas in Transformers interactions are mediated via multi-head dot product attention. The multi-head dot product attention mechanism (MHDPA) runs through the scaled dot-product attention multiple times in parallel. There is an important difference with NPS: in MHDPA, one can treat different heads as different rule applications. Each head (or rule) considers all the other entities as relevant arguments as compared to the sparse selection of arguments in NPS. We consider 8 rules, and at each position we apply 4 rules (each rule as a separate head).
> >
> > **Results:** Vision Transformer gets (71.1 +/- 2.1, 19.2 +/- 3.6, 18.9 +/- 2.5) accuracy for (Interpolation, Extrapolation, Held-out triples) (Higher is Better). Proposed architecture (with NPS as interactions) gets (77.6 +/- 1.9 , 28.7 +/- 1.1,  26 +/- 4.2). The hypothesis is that a model which is learning about the data generating distribution should be able to infer and reason about the underlying rule based structure, and hence should be more likely to be successful.
> >
> > We hope that as a preliminary experiment this satisfies the concern of the reviewer. Future work should investigate how to best integrate the ideas in the proposed work with Transformers.

---

> > > ### Author Response · Authors · 2021-08-27
> > > **Visualizing rules for bouncing balls**
> > >
> > > We further conduct an analysis of how rules are selected for the bouncing balls task in the paper to better understand the rules learned by NPS. The figure for this analysis can be found here: https://upload.vaa.red/D3rZi#a6138cc7355684ff4d2495495036b405 (shared anonymously). In the figure, we use an NPS model with 3 rules and 3 rule application steps. We analyze the entity selection and the rule selection as asked by the reviewer. A rule application on a slot is shown by highlighting that slot with blue color. The index of the applied rule is also mentioned in the slot. We can see that whenever a rule is applied on the slot representing the background, rule 0 is used. On the other hand, whenever it is applied on the slots representing one of the balls, rule 1 or rule 2 are used. We can also see that rules are mainly only being applied to the two balls in the middle that are touching or close to touching while no rules are being applied to the ball on the top since it stays constant throughout this episode. The ball at the bottom is also mostly constant and receives only 1 rule application when it's close to colliding with the wall. This shows that NPS learns rules that are interpretable in visually complex environments like bouncing balls as well.

---

> ### Author Response · Authors · 2021-08-30
> **Updated Impression?**
>
> Dear. Reviewer,
>
> Thanks for your time in providing feedback.
>
> **The potential impact for differentiable reasoning is enormous**
>
> Thanks again for recognizing that the potential is enormous.
>
> We think that your feedback has helped us to improve the presentation of our work. We also ran extra experiments as suggested by the reviewer (integrate NPS in a transformer, and performed experiments on RPM benchmark).
>
> More specifically, other concerns raised by the reviewer were also raised by another reviewer CM2F. Reviewer CM2F feels satisfied by our response (and also increased their score) regarding the difference between the proposed method and the baseline after performing thorough experiments and averaging over many seeds. Reviewer CM2F also raised concerns about the baselines used and we further clarified them using the below explanation which reviewer CM2F deemed convincing, we hope that this will also clarify any concerns that the reviewer has:
>
> We would like to further clarify our choice of baselines. In the paper, we have tested NPS on physical reasoning tasks. Any model applied to physical reasoning tasks has 3 components:
>
> - Representation of different entities i.e. how the raw observations are encoded into slots.
> - Individual dynamics of each entity.
> - Interaction among entities.
>
> Our contribution, NPS, is a contribution to the 3rd component i.e. it is a new kind of interaction network. For the first 2 components, we use the models already available in the literature and those that are prominently used. We refer to the first 2 components as the “backbone”. Since our model is an interaction network, we compare it to other interaction networks used in literature. These include GNNs [1, 2] and Multi-head attention [3, 4]. In every experiment that we perform, we keep the backbone constant and only vary the interaction network (i.e. change it to NPS). In each of our experiments, we show that NPS outperforms other interaction networks while keeping the backbone constant. For example, for bouncing balls, we compare against SCOFF [4]. In the case of SCOFF, we keep the convolutional encoder and the self-dynamics model, which in the case of SCOFF is an RNN, constant and only replace the multi-head attention in SCOFF with NPS. We follow the same procedure for each of our experiments. To ensure that the success of NPS does not depend on any particular backbone, we test it in a number of different backbones which include the OP3 backbone (object masking encoder) for spmot experiments, the RPIN backbone (convolutional encoder followed by convolution self-dynamics model) for shapes stack experiments, SCOFF backbones (slot attention encoder followed by recurrent self-dynamics model) for bouncing balls, etc. In each of these NPS outperforms the default interaction networks used. In conclusion, the only relevant baselines for NPS are other interaction networks. In deep learning literature, multi-head attention and Graph Neural Networks (interaction networks [5] are similar to GNNs) are generally used as interaction networks.
>
> Since the review period is coming to an end, we want to seek feedback on the experiments performed by us. We would also be happy to clarify any further concerns which reviewer may have.
>
> Thanks for your time, and help. We appreciate it. :)

---

### Decision · Program_Chairs · 2021-09-27

**Decision:**

Accept (Poster)

**Comment:**

This paper introduces a differentiable and learnable version of the classic production system architecture. This is a worthwhile and interesting attempt at enabling more flexible reasoning in deep learning systems. The reviewers found the initial evaluations of the system uncompelling, however the substantial additions provided during the discussion phase were deemed sufficient by all reviewers. This means the paper is an unusual case where it will need substantial revision to the results, but the reviewers believe this is feasible -- so I too will recommend acceptance. I urge the authors to also attempt to clarify the model presentation, as indicated by the reviewers.